# Superoxide Radicals in the Execution of Cell Death

**DOI:** 10.3390/antiox11030501

**Published:** 2022-03-04

**Authors:** Junichi Fujii, Takujiro Homma, Tsukasa Osaki

**Affiliations:** Department of Biochemistry and Molecular Biology, Graduate School of Medical Science, Yamagata University, Yamagata 990-9585, Japan; tkhomma@med.id.yamagata-u.ac.jp (T.H.); tosaki@med.id.yamagata-u.ac.jp (T.O.)

**Keywords:** superoxide, ferroptosis, radical electron, nitric oxide

## Abstract

Superoxide is a primary oxygen radical that is produced when an oxygen molecule receives one electron. Superoxide dismutase (SOD) plays a primary role in the cellular defense against an oxidative insult by ROS. However, the resulting hydrogen peroxide is still reactive and, in the presence of free ferrous iron, may produce hydroxyl radicals and exacerbate diseases. Polyunsaturated fatty acids are the preferred target of hydroxyl radicals. Ferroptosis, a type of necrotic cell death induced by lipid peroxides in the presence of free iron, has attracted considerable interest because of its role in the pathogenesis of many diseases. Radical electrons, namely those released from mitochondrial electron transfer complexes, and those produced by enzymatic reactions, such as lipoxygenases, appear to cause lipid peroxidation. While GPX4 is the most potent anti-ferroptotic enzyme that is known to reduce lipid peroxides to alcohols, other antioxidative enzymes are also indirectly involved in protection against ferroptosis. Moreover, several low molecular weight compounds that include α-tocopherol, ascorbate, and nitric oxide also efficiently neutralize radical electrons, thereby suppressing ferroptosis. The removal of radical electrons in the early stages is of primary importance in protecting against ferroptosis and other diseases that are related to oxidative stress.

## 1. Introduction

The free energy that is released from organic compounds by oxidation supports animal life. While the production of reactive oxygen species (ROS) increases during periods of elevated cellular metabolism in response to neuronal and humoral stimuli, their production further increases under pathological conditions, such as inflammation and ischemia–reperfusion injury. Superoxide is a primary radical that is produced when an oxygen molecule receives one electron through enzymatic or non-enzymatic reactions [1]. Superoxide is easily converted to other ROS, making it difficult to evaluate its contribution to an individual etiology. Because the radical electron can trigger chain reactions, superoxide dismutase (SOD) undoubtedly plays pivotal roles in the protection against ROS-mediated oxidative damage at the initial stage.

Excessive intracellular ROS concentrations tend to result in the oxidative modification of vital biomolecules, which can lead to accelerated ageing and the aggravation of certain disease processes. Moderate amounts of ROS, notably hydrogen peroxide, modulate cellular signaling through oxidative modification of target molecules in a reversible manner [2,3]. Accordingly, while antioxidant compounds or enzymes can protect against oxidative damage, the excessive elimination of ROS may impair cellular functions via disrupting redox signaling [4]. For these reasons, not only high levels of ROS but also an imbalance between ROS and antioxidants may impair redox homeostasis, leading to health problems [5].

Nitric oxide (NO), another radical, is also abundantly produced by both enzymatic and non-enzymatic processes and acts as an intra- and inter-cellular signaling molecule [6,7]. NO reacts with superoxide in a diffusion-limited manner. Although the resulting peroxynitrite (ONOO^−^) is often considered to be a toxic compound [8], from another point of view, this reaction terminates the chain reaction that is initiated by superoxide. Our body is protected by a wide range of antioxidative enzymes and compounds that coordinately act in antioxidation reactions, and their insufficiency enhances the toxic effects of ROS.

In this review article, we first revisit the representative sources of superoxide and ROS and briefly summarize the antioxidative enzymes that maintain ROS at adequate levels. We then survey the literature from viewpoint of free radical biology to explain how ROS are involved in cell death by focusing on ferroptosis, which is a recently identified free-iron dependent necrotic cell death [9]. We finally discuss the significance of eliminating superoxide by the action of antioxidative compounds, notably NO and ascorbate (Asc) to maintain redox homeostasis in the mammalian body.

## 2. Sources and Actions of Superoxide and Related ROS

Superoxide is a primary ROS that is produced via multiple enzymatic and non-enzymatic reactions that use oxygen molecules as an electron acceptor [1]. Enzymatic pathways for superoxide production are classified into two categories. One class of enzymes produces superoxide and, in some cases, hydrogen peroxide or other organic peroxides as principle products with oxygen consumption in a stoichiometric manner. The other class occasionally produces superoxide as a byproduct, a process that depends on environmental conditions (Figure 1). In the next paragraphs, we provide an overview on the major reactions that contribute to the production of superoxide and other ROS and, under certain circumstances, that cause oxidative damage to vital cellular biomolecules.

### 2.1. Mitochondrial Respiratory Chain

During carbohydrate metabolism via glycolysis and the tri-carboxylic acid (TCA) cycle, abstracted high-energy electrons are accepted in the forms of NADH and FADH_2_. In the mitochondrial electron transport chain (ETC), free energy is gradually released from the electron. ETC-I (NADH-ubiquinone oxidoreductase), ETC-III (cytochrome bc complex, ubiquinol-cytochrome c reductase), and ETC-IV (cytochrome c oxidase) act as proton pumps by utilizing energy from abstracted electrons and raise the elecrochemical potential across the inner mitochondrial membrane. The electrons are eventually converted to water by means of respired oxygen molecules at ETC-IV. ETC-V (F_o_F_1_-ATPase) is the molecular turbine that synthesizes ATP by employing the electrochemical potential that is produced.

The ETC eventually consumes more than 95% of the respired oxygen [10,11,12]. When an electron is mistakenly transferred to an oxygen molecule during these processes, superoxide anions are produced. This electron leakage makes mitochondrial ETCs the major sources of superoxide [12]. An ischemic situation is accompanied by a shortage of oxygen molecules, which causes electrons to be retained in the respiratory chain. Upon re-oxygenation, the reflowing oxygen molecules tend to accept these electrons and become superoxide before being converted into water at ETC-IV. Thus ischemia–reperfusion typically enhances the production of mitochondrial superoxide as well as that from other sources. Recent studies have indicated that several other carbon-metabolizing processes in the TCA cycle also can serve as a source for the production of superoxide/ROS under pathological conditions [13].

### 2.2. NADPH Oxidases

ROS are produced not only inside cells but also extracellularly, the latter of which is largely produced via membrane oxidases that utilize electrons from NADPH. Five members of NADPH-dependent oxidases (NOX) and two related oxidases (DUOX) convert molecular oxygen to superoxide and hydrogen peroxide as the primary products [14]. The produced ROS can then be released to the opposite side of the cytoplasm where they are utilized as bactericidal agents or move into the cytoplasm where they can modulate cellular signaling. Excessive ROS, however, can oxidatively modify cellular components and result in their dysfunction or the aberrant proliferation of cells [15]. Thus, dysregulation of the NOX/DUOX family of proteins is associated with inflammation and tumor development. The following are typical examples of reactions that are catalyzed by the NOX/DUOX family.

NOX2 is dominantly expressed in phagocytes and produces a relatively large amount of superoxide for the purpose of killing bacteria inside phagosomes in an inflamed area [14,16]. NOX2 proteins are constitutively present, and upon inflammatory stimuli, the activated NOX2 converts molecular oxygen into superoxide using electrons from NADPH and releases superoxide. The released superoxide reacts with other ROS and becomes a stronger bactericidal agent, such as hypochlorous acid, inside the phagosome [17]. A defect in phagocytic NOX2 causes chronic granulomatous disease (CGD), a syndrome that includes an enhanced susceptibility to specific microbial infections. Because the production of ROS by NOX2 easily exceeds cellular tolerance, they, together with inflammatory cytokines and eicosanoids, may exert adverse effects on surrounding tissues.

Accumulating evidence indicates that ROS, especially hydrogen peroxide, are produced in response to mitotic stimuli and act as signal modulators [2]. Pioneering work has been carried out on signaling of the receptor for platelet-derived growth factor (PDGF) that stimulates NOX1 in vascular smooth muscle cells and induces their growth [18]. It has also been reported that superoxide produced by NOX1 is involved in angiotensin II signaling [19]. We now know that a similar mechanism operates in many other processes, including receptor signaling for hormones and immune systems [20].

Many mitotic signals within cells are transduced by means of protein phosphorylation, notably the phosphorylation of tyrosine at the initial phase. While protein kinases phosphorylate signaling molecules and transmit signals, protein phosphatases dephosphorylate substrates and terminate this signal transduction. Different from phosphatases that dephosphorylate serine/threonine, phosphotyrosine phosphatases (PTP) commonly possess a cysteine (Cys) residue at the catalytic center and are sensitive to ROS-mediated oxidation. PTPs are transiently inactivated during an oxidative insult, which results in sustaining the phosphorylation of signaling molecules. While NOX/DUOX are mainly localized at the plasma membrane, they are occasionally present in the lumen of the endoplasmic reticulum (ER) or in endosomes [21]. NOX4 is a unique isoform that primarily produces hydrogen peroxide, which appears to act largely as a signaling molecule [22]. When ROS are released from NOX on the cell surface, only a portion of them may enter cells and affect signal transduction. This problem in efficacy can be eliminated by forming a vesicular structure, referred to as a redoxosome, which is formed upon ligand stimulation and enables ROS to more efficiently target PTP for the corresponding receptor [23]. PTP is a sensitive target of ROS, dominantly hydrogen peroxide, because the sulfhydryl group of the Cys residue in the catalytic center has a low pKa due to the structural microenvironment and, hence, tends to form a Cys thiolate anion (Cys–S^−^) [24]. Hydrogen peroxide preferentially oxidizes Cys–S^−^ to Cys–sulfenic acid (Cys–SOH), which transiently inactivates PTP and prolongs mitotic signals. Cys–SOH can eventually be reduced back to Cys–SH by other reducing molecules.

Due to the fact that ROS indiscriminately target biomolecules, they may also impair cellular functions. For example, while a small amount of ROS may stimulate insulin receptor signaling, ROS generally cause insulin tolerance [25]. It is now well established that unregulated ROS signaling is the cause of aberrant cell growth and eventual tumor development or cellular senescence [26]. Excessive hydrogen peroxide causes hyperoxidation of the catalytic Cys and, beyond Cys–OH, results in the formation of Cys–sulfinic acid (Cys–SO_2_H) and further oxidized Cys–sulfonic acid (Cys–SO_3_H). Cells do not have redox systems for reducing these hyperoxidized Cys species back to Cys–SH, such that PTP with a hyperoxidized Cys is permanently inactive. As a result, the phosphorylation signal is transmitted excessively and may accelerate tumor development as the result of uncontrollable cell proliferation. The association of inactivated PTPs with a variety of tumors has been reported [27]. The same signal-modulatory mechanism can also be seen in other Cys–centered enzymes. Phosphatase and tensin homolog deleted from chromosome 10 (PTEN) dephosphorylates phosphatidylinositol (3,4,5)-trisphosphate (PIP_3_) to phosphatidylinositol (4,5)-bisphosphate (PIP_2_) and consequently suppresses the PI3K/Akt signaling pathway. Either a mutation or hyperoxidation causes PTEN to be inactive, leading to the acceleration in tumor development [28]. As a result, PTEN is one of the most important tumor-suppressor gene products. To prevent inactivation of these phosphatases, several defensive systems are available, and these function to maintain ROS signaling in an appropriate range. As detailed below, peroxiredoxins (PRDXs), which exhibit thioredoxin (TRX)-dependent peroxidase activity, have a regulatory role in restricting hydrogen peroxide levels, thereby maintaining normal ROS signaling [3,29].

Cell cycle regulation is another example of NOX-involved signaling, although the dephosphorylation of corresponding molecules is involved in advancing the cell cycle. Among the phosphatases, Cdc25 is a dual phosphatase and a master regulator of the cell cycle and stimulates CDK activity via the dephosphorylation of cyclin/CDK complexes [30]. Similar to PTP, Cdc25 possesses a reactive Cys unit at the catalytic center. ROS activates Cdc25 via enhanced dephosphorylation or transiently inactivates it by the hyperoxidation of Cys in the active site [31,32]. NOX4-derived ROS are reportedly required to induce the dephosphorylation of Cdc25c, which then activates cyclin/CDK complexes and promotes the cell cycle in melanoma cells [33]. Conversely, Cyclin B1/Cdk1, which is partly localized to the mitochondrial matrix, phosphorylates some mitochondrial proteins including ETC-I and is involved in increasing the energy supply for the G_2_/M transition [34]. Findings reported in a recent study employing specific chemical probes suggest that ROS levels peak in mitosis, and that the oxidation of proteins and DNA occurs at the G_2_/M phase [35]. When in excess, ROS appear to arrest cell growth and occasionally cause cell death, which may be the mechanism for the suppression of tumorigenic cell growth by anti-cancer drugs [3]. However, the issue of how ROS production is regulated during the cell cycle is still unclear.

### 2.3. Cytochrome P_450_ (CYP)/Cytochrome P_450_ Reductase (POR) System

The reaction of CYP coupled with POR releases superoxide as a byproduct of the oxidase reaction [36]. A number of hydrophobic compounds that are either intrinsically produced, or xenobiotics including toxicants and drugs, are conjugated with sulfate, glucuronate and glutathione for the purpose of accelerated excretion. CYP acts in the first phase of these detoxification reactions and introduces an oxygen-containing group, making them susceptible to being conjugated. POR transfers an electron from NADPH to CYP and enables the continuous oxygenation of substrate compounds [37]. The reaction of heme oxygenase HO-1 and HO-2, which oxidatively degrades heme, also accepts electrons from NADPH via POR. ROS that are produced by CYP, which are associated with the ER membrane and are partly localized in mitochondria, occasionally trigger oxidative damage [38,39]. The liver is the main organ for the detoxification of drugs or xenobiotics and is therefore prone to damage due to them being present in excess, which may be associated with non-alcoholic liver diseases [40]. CYPs are also involved in steroidogenesis in the gonads and produce ROS as byproducts. Hence, testes in adult males and ovaries, notably in pregnant females, are exposed to excessive ROS released by CYPs during steroidogenesis.

### 2.4. Xanthine Oxidoreductase

Xanthine oxidoreductase (XOR) is an enzyme that contains a molybdenum atom as a cofactor and catalyzes the oxidative conversion of purine bases to uric acid. Xanthine dehydrogenase (XDH) activity of XOR abstracts electrons from purine bases and donates them to acceptor molecules (NAD^+^), and the resultant electrons may be utilized in biological reactions such as mitochondrial ATP synthesis. The oxidation of specific sulfhydryl groups of Cys residues in or limited proteolysis of XDH converts XDH into xanthine oxidase (XO) [41]. This transition appears to involve a thermodynamic equilibrium between XDH and XO [42]. The resulting XO utilizes oxygen molecules, instead of NAD^+^, as the electron acceptor and produces superoxide or hydrogen peroxide. Hence, along with other systems such as NOX and mitochondrial ETC, it has been proposed that XO is involved in ischemia–reperfusion injury [43]. Moreover, XOR participates in additional reactions that dominantly include nitrite reduction in cardiovascular system [44]. The resulting NO is involved in vasodilation, leading to an increase in blood flow. Because the excessive accumulation of uric acid in the blood may cause gout, XOR is a well-established target of drugs that treat this condition. However, considering the multiple physiologic action of XOR, we need to pay more attention to methods for inhibiting the action of XOR for the purpose of treating of gout.

### 2.5. Non-Enzymatic Production of Superoxide

Superoxide can also be produced by several non-enzymatic reactions. Above all, non-enzymatic glycosylation, designated as glycation, proceeds under conditions of hyperglycemia and produces a variety of compounds [45,46]. Glycoxidation, which proceeds during glycation, largely produces ROS, including superoxide, and appears to play roles in diabetic complications, neurodegenerative diseases and cancer [47]. It is thought that the amount of ROS produced by this process under hyperglycemic conditions may actually be equal to the amount produced by the enzymatic reaction. Accordingly, their lifetime production may cause persistent damage to the body.

## 3. Antioxidative Enzymes and Compounds to Protect against ROS-Induced Oxidative Damage

The oxidizing power of superoxide itself is moderate compared with the hydroxyl radical. However, superoxide is the primary radical that either provides an electron to or abstracts one from other compounds, which results in the production of other radical species via a chain reaction until the radical electron is eventually eliminated. The resulting radicals and other ROS that are produced during chain reactions can trigger oxidative injury. Hence, superoxide itself may not directly cause severe damage, but if not removed properly, it potentially leads to cell damage and death.

### 3.1. Dismutation of Superoxide to Hydrogen Peroxide Abolishes Radical Electrons

The conversion of superoxide to hydrogen peroxide and an oxygen molecule is a type of dismutation that occurs spontaneously (rate constant k = 5 × 10^5^ M^−1^s^−1^ at neutral pH). SOD greatly accelerates this reaction, approaching diffusion-limited access (k = 1.6 × 10^9^ M^−1^s^−1^) under a broad pH range, pH 5.3–9.5 in the bovine enzyme [1]. This high catalytic ability is enabled partly by electrostatic interactions in the active center of the SOD protein, which was revealed by the structural analysis of it [48,49]. The evolutionary acquisition of SOD appears to be advantageous for most organisms because the ancestor for mammalian SODs, Fe-containing SOD, was present even when atmospheric oxygen levels were much lower than today [50]. Mammals produce three forms: SOD1, SOD2 and SOD3, which are first found in erythrocytes, muscle, and lung, respectively, and are now known to be present fairly abundantly in the body. Because details in genetic and evolutionary aspects of SODs can be found elsewhere [51], we here outline their profiles from aspects of regulation of their activities.

#### 3.1.1. SOD1

SOD1, which is a copper (Cu)- and zinc (Zn)-containing enzyme, is the most abundant among the three isoforms in the body and is localized in the cytosol and partly in intermembrane space of the mitochondria [52]. SOD1 largely consists of a beta barrel structure [49], which stabilizes the three-dimensional structure of the protein and makes it resistant to heat and other types of environmental stress. SOD1 is stored in the stable apo-form and can be activated by the posttranslational insertion of copper, which is mediated by a copper chaperone for SOD1 (CCS) [53]. CCS^−/−^ mice show decreased SOD1 activity, but a significant amount of activity remains, which suggests the presence of a CCS-independent activation pathway [54]. Based on studies of yeast and other lower organisms, the existence of the presence of a CCS-independent pathway has been proposed for the maturation of SOD1 that involves the insertion of Cu [55]. In the case of mitochondrial SOD1, the protein is present as an inactive form but is activated upon exposure to low concentrations of hydrogen peroxide, independent from the Cu status of the organism [56]. Mitochondrial ETC and thioredoxin reductase appear to be responsible for the activation of SOD1 in cells, suggesting the existence of a link to mitochondrial energy metabolism [57].

While a variety of diseases are reportedly associated with altered SOD1 activity, the following two hereditary diseases have attracted much attention. Because a trisomy of the 21st chromosome is the cause for Down’s syndrome and SOD1 is mapped to the region where the putative causative gene is thought to be present, an elevation in SOD1 activity has been the suspected cause of Down syndrome [58]. Pathological abnormalities in the neuromuscular junctions in the tongue are characteristically observed in transgenic mice overexpressing SOD1 [59]. According to this hypothesis, it is conceivable that elevated levels of hydrogen peroxide due to increased SOD1 cause the generation of hydroxyl radicals by the Fenton reaction, thereby causing neuronal damage. This hypothetical mechanism, however, does not explain an important issue regarding the radical source required for the reduction of ferric iron to ferrous iron. SOD family enzymes consume two superoxide radicals quite rapidly, which would appear to decrease the chance that superoxide is the electron source for the eventual production of ferrous iron. Thus, this hypothetical mechanism should be revised based on our current knowledge of free radical biology.

SOD1 is the first gene in which missense mutations have been reported for familial amyotrophic lateral sclerosis (FALS) [60]. As of this writing, more than 150 mutations have been identified in the SOD1 gene, and the presence of mutant SOD1 proteins has been confirmed in these patients [61]. Several mouse models have been established by the transgenic overexpression of the mutant human gene and have been employed in attempts to elucidate the etiology of ALS and to develop possible treatments [62]. SOD1 activity in these mice is commonly higher than in wild-type mice because of the presence of the normal mouse SOD1 allele, while FALS patients with the SOD1 mutation commonly exhibit decreased activity. In addition to SOD1, mutations have been found in other genes in FALS patients, although they show little association with antioxidation.

The establishment of the first SOD1-knockout mouse was reported in 1996 [63], followed by two other groups independently in 1998 [64,65]. It was surprising that the genetic ablation of SOD1 do not induce ALS in any of these mice and that they exhibit only moderate phenotypic abnormalities. Given the results for transgenic mice with excessive SOD1 activity derived from the transgene, it is conceivable that the loss of SOD1 function is not the direct cause for ALS. This notion is consistent with findings that human subjects with a loss-of-SOD1-function mutations do not develop ALS but other phenotypic abnormalities [66,67]. While CCS supports the efficient incorporation of copper into SOD1 in motor neurons, the CCS deficiency does not modify the onset or progression of neuron disease in transgenic mice overexpressing mutant SOD1 [68]. Collectively, this evidence suggests that the gain-of-function of mutant SOD1 is the actual cause for the emergence of ALS. However, there still remains ambiguities regarding the mechanism responsible for how mutant SOD1 causes FALS, and studies on this issue are now underway from the perspective of basic and clinical medicine.

#### 3.1.2. SOD2

SOD2 is a manganese (Mn)-containing isozyme that appears to have evolved from the same ancestral gene as the current bacterial iron-containing SOD and is exclusively localized in the mitochondrial matrix [69]. While SOD1 can be activated by binding copper post-translationally, Mn is inserted into SOD2 co-translationally [70]. The SOD2 gene is localized in the nucleus, such that SOD2 is synthesized with an *N*-terminal leader peptide that is required for mitochondrial import. The leader peptide is cleaved off after exposure to the mitochondrial matrix. A membrane transporter, SMF2, has been identified as a mitochondrial carrier protein for Mn [71]. Both SOD2 and catalase are induced by a transcriptional regulator FoxO under conditions of oxidative stress and appear to exert antioxidative properties and function against senescence [72]. Restricted calorie intake is a promising intervention in extending the life span of mammals by reducing oxidative stress and damage. Sirtuin has NAD^+^-dependent deacetylase activity and its levels are increased by caloric restriction. A mitochondrial sirtuin SIRT3 deacetylates two critical lysine residues on SOD2, promotes its antioxidative activity, and reduces ROS [73]. However, the overexpression of SOD2 alone does not extend the lifespan in mice [74].

SOD2^−/−^ mice die within 10 days after birth with dilated cardiomyopathy [75]. This observation is consistent with results on the transgenic overexpression of SOD2 that protects mice against ischemia–reperfusion injury to the heart [76]. Subjects with a cardiomyocyte-specific SOD2 deficiency can reach adolescence but die at around 4 months of age due to heart failure [77]. While skeletal muscle cells also contain sufficient mitochondria to fulfill the requirement of ATP for contraction, the heart is continuously contracting, which results in the continuous production of superoxide. The accumulation of lipid in the liver and skeletal muscle is typical in SOD2-deficient mice, and this would be caused by impaired mitochondrial β-oxidation [75,78]. SOD1-deficient mice under conditions of starvation also show an accumulation of lipid droplets systemically [79,80] in which mitochondrial dysfunction may also be involved. Although SOD2^+/−^ mice appear to be healthy, interventions such as hyperoxygen and pertussis toxin increase their susceptibility [81,82]. A heterozygous deficiency of either SOD1 or SOD2 causes the activation of the mitogenic signaling pathway in aortic smooth muscle cells [83]. Mitochondria isolated from SOD2^−/−^ mice livers consistently show increases in the levels of oxidative stress markers, such as 8-hydroxyguanine and carbonyl proteins, and a concomitant decrease in Fe-S proteins, while no significant damage have been found in cytoplasmic proteins and nuclear DNA [84]. Unlike SOD1^+/−^ mice, SOD2^+/−^ mice show an impaired recovery from postischemic failure of contraction. SOD2^−/−^ mice also show increases in superoxide levels and vascular dysfunction compared to control mice [85]. The conditional knockout mice show increases in ROS followed by the production of 4-hydroxynonenal inside mitochondria. Thus, the ablation of SOD2 generally shows more severe phenotypic abnormalities than that of SOD1, suggesting the pivotal role of this protein in eliminating mitochondrial superoxide for the longevity of mice.

#### 3.1.3. SOD3

SOD3 is a secreted form and contains sugar chains that increase the stability of this protein in the extracellular milieu [86]. SOD3 is also a Cu- and Zn-containing enzyme whose core structure is similar to that for SOD1 but is translated with amino-terminal signal sequences that are required for secretion from cells. SOD3 also has a carboxyl terminal domain that is rich in positively charged amino acids, which enables it to bind to heparin sulfate on the endothelial cell surface in blood vessels [87]. Lungs contain high levels of SOD3, where it is considered to play pivotal roles in the pulmonary system that is exposed to hyperoxygen conditions in the body. SOD3 specifically inhibits dendritic cell maturation and regulates adaptive immune responses, which results in maintaining lung homeostasis [88]. The mechanism responsible for the assembly of active SOD3 is becoming clear. The copper chaperone antioxidant-1 (Atox1) is involved in delivering copper to SOD3 at the trans-Golgi network by interacting with the copper transporter ATP7A [87,89,90]. Atox1 shows similarity to the *N*-terminal domain of CCS and also serves as a transcription factor for inducing SOD3 [91].

SOD3 is also expressed at high levels in the vascular system [92]. The overexpression of SOD3 actually protects the vascular system from oxidative injury [93,94], and protects the kidney from ischemia/reperfusion injury [95]. Regarding its function in the vascular system, the SOD3-mediated elimination of superoxide preserves NO and the concomitant suppression of ONOO^−^ formation [87,96], which results in protecting the heart from oxidative stress and suppressing myocardial infarction [97]. Thus, SOD3 is regarded as a potent antioxidant, notably in the lungs and cardiovascular systems. SOD3 also suppresses skin inflammation and innate immune responses during bacterial infections [98,99], thereby suppressing fibrosis, but the mechanism differs from that of the immune system-involved reaction [100].

In an initial study, SOD3-deficient mice showed moderate phenotypes, such as elevated levels of lipid peroxidation products and decreased ascorbic acid levels in the plasma [101]. Analyses of the development of SOD3-deficient mice have unveiled potential roles of the gene, most notably in the lungs, the cardiovascular system, and the skin [102,103]. Since the loss of SOD3 by conditional knockout leads to severe lung damage [104], SOD3 is thought to play a role in the survival of cells in the presence of ambient oxygen. Chronic obstructive pulmonary disease (COPD) is a major public health problem worldwide, and fragmentation of the extracellular matrix is triggered by oxidative stress and participates in the development of COPD. A statistically significant association between SOD3 variants and COPD susceptibility has been reported [105,106]. Moreover, SOD3 has actually been found to attenuate emphysema and reduce the fragmentation of the extracellular matrix via its antioxidative mechanism [107].

### 3.2. Peroxidases Complete Radical Detoxification

Hydrogen peroxide, which is caused by either the spontaneous dismutation of superoxide or the catalytic action of SOD, is still a reactive species but is also a relatively stable ROS. Moreover, a variety of peroxides are produced during physiological processes. The presence of free transient metals, especially ferrous iron, may lead to the production of hydroxyl radicals via Fenton chemistry [108]. Hence, enzymes with peroxidase activities, which include catalase, glutathione peroxidase (GPX), and PRDX, are present in abundance and restrict hydrogen peroxide and other peroxides to manageable levels.

#### 3.2.1. Catalase

Catalase contains heme and is ubiquitously present in peroxisomes of eukaryotic cells, where it catalyzes the dismutation of two hydrogen peroxide molecules to water and an oxygen molecule. The reaction is quite fast, but the Km for hydrogen peroxide is high. Acatalasemia, the inherited deficiency of catalase activity, has been considered to be a benign phenotype. Further studies, however, imply that patients or catalase-null mice actually exhibit a complicating condition for aging and oxidative damage [109]. Peroxisomes are the organelles where some types of phospholipids are synthesized. Because polyunsaturated fatty acids (PUFA) are targets of lipid peroxidation by ferroptotic stimuli, the lipogenic activity of peroxisomes may affect the sensitivity of cells to ferroptosis [110]. Peroxisome proliferator-activated receptors (PPARs) are members of the nuclear hormone receptor superfamily and are involved in many cellular processes, such as proliferation, death, and differentiation via the regulation of lipid and carbohydrate metabolism [111]. Activation of PPARδ has been found to induce catalase expression, which renders xCT-deficient mouse embryonic fibroblasts resistant to ferroptosis [112]. Because xCT deficiency involves decreased glutathione (GSH) synthesis due to an insufficient uptake of Cys, thus making GPX less potent, catalase may support cell viability by suppressing Fenton chemistry, which would result in resistance to ferroptosis.

#### 3.2.2. GSH-GPX System

Different from catalase, the reductive detoxification of peroxides by GPX requires GSH for the electron donor and results in the formation of an oxidized glutathione dimer that is linked by a disulfide bridge (GSSG). Cellular GSH is maintained by a balance between de novo synthesis and the reductive recycling of GSSG by glutathione reductase (GSR) in a NADPH-dependent manner [113]. γ-Glutamylcysteine synthetase (γ-GCS) combines Cys and glutamate (Glu) to produce γ-Glu-Cys. GSH is produced when γ-Glu-Cys is utilized for the glutathione synthetase (GSS) reaction, which adds glycine (Gly) to the carboxyl end of γ-Glu-Cys.

While Cys is a semi-essential amino acid (present in sub-millimolar concentrations in the liver) and plays a central role in GSH (present in several millimolar concentrations in the liver) function, there is a tendency for Cys to be insufficient because it is also a precursor for a variety of indispensable compounds, such as taurine, cysteamine, and coenzyme A, in addition to a constituent for proteins and GSH [114]. Buthionine sulfoximine is a potent inhibitor of γ-GCS and lowers cellular GSH levels [115] but does not cause a decrease in intracellular Cys levels or induces ferroptosis in some types of cells during the short-term use of the inhibitor [116]. This is in sharp contrast to a dysfunction of xCT, a cystine–glutamate antiporter encoded by SLC7A11; the inhibition of xCT results in the prompt decline in cellular Cys levels and rapidly leads to the development of robust ferroptosis [9]. While the anti-ferroptotic action of GSH is solely attributed to the donation of an electron to GPX4, free intracellular Cys appears to exert an additional function in terms of protection against ferroptosis, although the overall process is not clearly understood at this time [117]. Several proteins and other compounds, such as vitamin E and nitric oxide, have been found to exert antiferroptotic activity independently of GPX4, as discussed below.

γ-GCS promiscuously develops a γ-glutamyl linkage with amino acids other than Cys, which results in a variety of γ-glutamylpeptides that are produced under Cys–insufficient conditions [115]. This trend is especially critical in the liver with an acetaminophen overdose [118], which causes the consumption of GSH for use in conjugation reactions and a subsequent Cys deficiency. The utilization of 2-aminobutyrate instead of Cys produces γ-glutamyl-2-aminobutylglycine (ophthalmate), such that ophthalmate is implied as a biomarker for acetaminophen-induced hepatotoxicity [119]. Despite extensive studies, the functions of the resulting γ-glutamylpeptides including ophthalmate are ambiguous. A recent study, however, provided information regarding a hypothetical role of γ-glutamylpeptides as well as the promiscuous reaction of γ-GCS [120]. Kang et al. [120] demonstrated that, under Cys insufficiency, excessive Glu exhibits cytotoxicity by activating mitochondrial metabolism and the production of ROS, which consequently causes ferroptosis. Given these observations, the formation of γ-glutamylpeptides by γ-GCS may decrease the levels of cellular Glu, thereby avoiding the risk of ferroptosis. However, there remains ambiguity concerning precisely how excessive Glu is involved in ROS production because Glu (present in several millimolar concentrations in the liver) is the most abundant free amino acid inside cells and plays a central role in the metabolism of other amino acids through aminotransferase reactions.

GPX constitutes a major peroxidase family that comprises eight members in mammals and utilizes GSH as the electron donor [121]. Selenocysteine (Sec), an amino acid with a structural similarity to Cys but contains selenium instead of sulfur, performs a central role in a series of proteins that are involved in the redox reactions. In mammals, GPX1 to GPX4 contain Sec at the catalytic center, but GPX5 to GPX8 contain Cys [122]. GPX reductively detoxifies, not only hydrogen peroxide, but also certain organic hydroperoxides, converting them into the corresponding alcohols. A recent study provided evidence for how a Cys/Sec residue in several enzymes with peroxidase activity can become peroxidatic (Cys–SOH or Sec–SeOH) [123]. Excessive hydrogen peroxide may further oxidize Sec–SeOH, resulting in the formation of dehydroalanine, which is a permanently inactive form [124]. Cys–substituted mutant GPXs are much less effective in the elimination of peroxides [122]. Cys–containing GPXs, GPX7 and GPX8 appear to be more important in catalyzing the oxidative folding of nascent proteins in the ER by utilizing the oxidizing power of peroxides [125].

The genetic ablation of GPX members shows mostly asymptomatic or benign phenotypes, but the ablation of GPX4 results in embryonic lethality [126,127]. This lethality can be overcome by vitamin E supplementation, suggesting the involvement of lipid peroxidation in this instance. These observations reveal the importance of the reduction of phospholipid hydroperoxides to the corresponding alcohol in rescuing cells from lethality [128]. The genetic ablation of GPX4 or the inhibition of GPX4 by chemical compounds such as RSL3 also induce ferroptosis [9]. A recent study employing mice with Cys–containing GPX4 revealed that Sec-containing GPX4 is more resistant than the Cys–containing variant to irreversible hyperoxidation and, hence, is more advantageous in terms of combating ferroptosis [129]. Accordingly, GPX4 has now attracted considerable interest and is discussed in the anti-ferroptosis section below. A detailed overview of GPX family is provided in a recent review article [130].

#### 3.2.3. TRX-PRDX System

TRX is a small redox protein and was originally identified as a subunit of ribonucleotide reductase. TRX has now been confirmed to be pivotal electron donor for multiple reactions and plays essential roles in maintaining redox homeostasis [131]. PRDX exhibits peroxidase activity toward hydrogen peroxide and other hydroperoxides primarily using TRX as an electron donor [132]. PRDX family proteins are classified into six members. PRDX1-4 are a class of dimeric proteins with similar structures, while PRDX5 and PRDX6 are monomeric enzymes that show more divergent structures. In addition to peroxidase activities, PRDX possess unique functions, including chaperon activity in protein folding, the modulation of cellular signaling, and oxidative protein folding in the ER.

This protein family contains highly reactive Cys residues, which form Cys–SOH by reactions with peroxides, and is converted into disulfides with Cys–SH in another subunit of homodimeric PRDX or within its own peptide molecule in monomeric PRDX [132]. The intra- or inter-molecular disulfide bond is then reduced back to Cys–SH by the action of TRX. Oxidized TRX is then reduced back by thioredoxin reductase, which is another member of the family of Sec-containing enzymes and employs NADPH as the primary electron donor [133]. In the presence of excessive levels of peroxides, Cys–SOH is hyperoxidized to cysteine sulfinic acid (Cys–SO_2_H) and then to cysteine sulfonic acid (Cys–SO_3_H). The resulting PRDX containing Cys–SO_2_H or Cys–SO_3_H is irreversibly inactivated under physiological conditions. However, there is an exception to the system in which the Cys–SO_2_H in PRDX1, 2 and 3 are converted back to Cys–SH, a reaction that is performed by sulfiredoxin in an ATP-dependent manner [134,135].

PRDX1 and PRDX2 are cytosolic isoforms and play roles in the modulation of phosphorylation signaling, notably for receptor tyrosine kinases, by controlling the amount of hydrogen peroxide to appropriate levels [3,29], while PRDX3 is the mitochondrial isoform [132]. PRDX4 is an ER-resident isoform and is dominantly involved in oxidative protein folding for secretory proteins, functioning in a similar manner to GPX7 and GPX8 [125]. PRDX5 is localized in several organelles and may be involved in cellular metabolism [136]. PRDX6 is one-Cys PRDX that exhibits peroxidase activity via the use of GSH as an electron donor [137,138]. GSH-dependent reduction of phospholipid hydroperoxide is an intrinsic function of PRDX6, although it is structurally dissimilar to GPX4 [139]. PRDX6 also exhibits calcium-independent phospholipase A2 (iPLA2)-like activity, which is differentially regulated from peroxidase activity [140] and may participate in membrane repair [141]. Thus, PRDX6 is a bifunctional enzyme and may also suppress ferroptosis via these unique properties as described below.

## 4. Cell Death Associated with an Oxidative Insult

ROS, when they are produced locally in response to stimuli, exert signaling actions, but, upon the production of excessive amounts, they lead to dysfunction and, in extreme cases, cell death. Apoptosis is the major type of cell death and is considered to be associated with approximately 90% of homeostatic tissue turnover [142]. Other mechanisms of non-apoptotic cell death are classified into several categories depending on types of cells and the signaling pathways involved [143]. Ferroptosis is a novel type of necrotic cell death in which free iron and lipid peroxides are intimately involved [9]. Here, we briefly discuss the concept of apoptosis in association with ROS and then discuss ferroptosis, mainly focusing on ambiguous issues associated with this oxidative stress-related cell death. Please refer to another review article, e.g., [144] for information on how ROS are involved in other types of cell death, as this is beyond the scope of this review article.

### 4.1. Roles of ROS in Apoptosis

Oxidative stress-involved cell death has long been regarded a subclass of apoptosis or, more simply, conventional necrosis. Apoptosis is principally the consequence of the activation of the caspase-mediated proteolytic pathway, as has been overviewed repeatedly in the literature, e.g., [145]. Fas is a death receptor that, upon stimulation by the Fas ligand or antibody binding, solely transduces apoptotic signals by means of the activation of caspase 8 followed by caspase 3. The activated caspase 3 then degrades the inhibitor for caspase-activated DNase (ICAD) [145]. The released caspase-activated DNase (CAD) is transported into the nucleus and cleaves chromatin, leading to the formation of a ladder-like structure of DNA, a hallmark of apoptosis. With nuclear condensation, apoptotic cells disintegrate into small apoptotic bodies. Although phosphatidylserine, which is localized in the inner leaflet of the plasma membrane in live cells, is exposed to the outer leaflet by means of scramblase and acts as the “eat-me signal”, the cell membrane remains intact and prevents the leakage of cellular contents. The Fas-mediated apoptotic pathway is unique, and ROS may not play an essential role in transducing the death signal.

A variety of stimuli that include ROS activate intrinsic pathways for apoptosis also appear to largely contribute to pathological cell death. In this pathway, cytochrome c is released from the stimulated mitochondria and associates with the protein platform Apaf-1, which then assembles apoptosomes and leads to the activation of caspase 9 [146]. Oxidized cytochrome c, but not the reduced form, is required for the assembly and activation of apoptosomes in the intrinsic pathway [147]. Relationships between cytochrome c oxidation and apoptosome function have now been examined in great detail [148]. Cardiolipin is a mitochondria-specific phospholipid that interacts with cytochrome c. The formation of a peroxidase complex between cytochrome c and cardiolipin stimulates the oxidation of cardiolipin [149]. While this peroxidase complex specifically uses cardiolipin as the substrate, the peroxidase activity appears to be involved in the execution of apoptosis. Following the formation of apoptosomes, caspase 9 proteolytically activates pro-caspase 3, which then follows the same processes as observed in the Fas-mediated apoptotic pathway.

Caspases are cysteine proteases in which a reactive Cys sulfhydryl (Cys–SH) residue is involved in the catalytic reaction. The dissociated form of sulfhydryl, the thiolate anion (Cys–S^−^), performs hydrolysis of target proteins, as is typically observed in PTP or other Cys–centered enzymes. Therefore, cysteine proteases, including caspases, are prone to oxidative inactivation in response to excessive levels of or strong ROS. In vitro studies indicate that singlet oxygen inactivates caspases [147] and Cys–centered lysosomal protease cathepsin B and L/S in cultivated cells [150]. Thus, ROS may be stimulants that induce apoptosis, but extreme oxidative stress appears to abort the apoptotic process by inhibiting the action of caspases. The oxidative inactivation of cysteine proteases actually occurs in in vivo situations. For example, caspase 1, which is not involved in apoptosis but is responsible for the proteolytic activation of interleukin 1 (IL-1) under inflammatory conditions, is oxidatively inactivated in the SOD1-knockout mouse treated with a lipopolysaccharide [151]. Surprisingly, this oxidative inactivation of caspase 1 causes the SOD1-knockout mouse to be less sensitive to lethality under conditions of immune stimulation.

### 4.2. Ferroptosis as a Radical-Associated Cell Death

While apoptosis is a clean type of cell death and affects surrounding cells only minimally, necrotic cells undergo rupture and release most of their cellular components, designated as damage-associated molecular patterns (DAMPs), which aggravates inflammatory responses and may cause a delay in healing and, eventually, autoimmune responses [152]. The necrotic cell death that is caused by elevated lipid peroxidation has been considered to occur in a non-specific manner. Polyunsaturated fatty acids (PUFA) are highly sensitive to oxidative modification by ROS, notably to hydroxyl radicals. Free iron, more specifically ferrous iron, can react with hydrogen peroxide, leading to the production of hydroxyl radicals. We now know that lipid peroxidation is not only a hallmark of oxidative stress but, in association with free iron, triggers necrotic cell death, a process that is referred to as ferroptosis [9,153]. Because lipid peroxidation occurs in a variety of pathogenic conditions with elevated ROS production, it is conceivable that ferroptosis could be involved in a variety of diseases that include the aggravation of ischemic diseases and neurodegenerative diseases [153]. For example, it has been suggested that ferroptosis is an aggravating factor for nephrotoxicity during folic acid-induced acute kidney injury [154]. Conversely, cancer malignancy appears to be associated with resistance to the ferroptosis that is induced by chemotherapy [108].

#### 4.2.1. Roles of Autophagy in Providing Free Iron

The free iron that is essential for causing ferroptosis might originate from both extracellular and intracellular sources [155,156]. Transferrin is the protein that binds iron in blood plasma and provides iron to cells via endocytosis. Most cancer cells require iron for rapid proliferation, while transferrin appears to be the source of the iron for inducing ferroptosis under Cys deficiency [156]. A large portion of iron is present as the heme-bound form, and free iron is mainly stored in cells as ferritin-bound ferric iron. Upon ferroptotic stimuli caused by a Cys deficiency, ferritin undergoes autophagic degradation, which is referred to as ferritinophagy, and becomes the source of the iron that is involved in lipid peroxidation reactions [143], which is schematically represented in Figure 2. This is represented by findings that the administration of lysosome inhibitors to cells that had been treated with erastin or RSL3 prevents the release of the free iron from ferritin into the cytoplasm, leading to an elevation in lysosomal ROS [157]. There are also several reports that confirm the involvement of ferritinophagy in providing iron for ferroptosis [158,159]. Nuclear receptor coactivator 4 (NCOA4) acts as a cargo receptor for the autophagic degradation of ferritin in ferroptosis. In the case of Cys deficiency, GPX4 undergoes degradation via chaperon-mediated autophagy, which is also associated with the execution of ferroptosis [160].

The lysosome is the organelle that contains high levels of α-tocopherol (Toc), approximately one-order of magnitude higher than other subcellular compartments [161], which may prevent lipid peroxidation of components of the lysosomal membrane. As long as the released ferric iron remains in the autolysosome, free iron would not trigger ferroptosis. Because the distribution of ferrous iron in the cytoplasm is a hallmark for ferroptosis, it would need to be released out of the autolysosome. GPX4 inactivation or a vitamin E deficiency may make the autolysosomal membrane fragile and result in its rupture. Then, radical species would extend lipid peroxidation reactions to the plasma membrane, which would then destroy the integrity of the cell membrane.

Heme iron is the most abundant form of cellular iron, but it is not redox reactive. Heme oxygenase (HO) is a unique enzyme that oxidatively degrades heme and releases free iron as well as carbon monoxide and biliverdin. Contradictory functions, either protective or injurious, have been reported for HO-1 in pathological conditions [162]. While erastin-induced ferroptosis is protected by HO-1 in renal proximal tubule cells [163], that in HT-1080 fibrosarcoma cells is stimulated by overexpressed HO-1 [164]. The autophagic degradation of mitochondria, designated as mitophagy, degrades macromolecules but also heme by means of HO incorporated into autolysosomes [164]. Although the stimuli for conventional mitophagy are different from those for ferritinophagy, the release of excess levels of ROS may damage mitochondria and result in their autophagic degradation. However, there is still an important issue that remains unsolved, namely, where do radical electrons come from?

#### 4.2.2. Involvement of Mitochondrial Superoxide in Ferroptosis

In addition to the presence of free iron, a continuous supply of radical electrons is required for the reduction of ferric iron to ferrous iron and continuing lipid peroxidation reactions. Several potential sources of free iron have been recently reported [165]. Many studies imply that mitochondria play pivotal roles in supplying radical electrons under conditions where ferroptosis is associated with metabolic processes, while oxygenases that involve PUFA peroxidation, notably arachidonate by arachidonate lipoxygenase (ALOX), produce lipid peroxides enzymatically which initiate ferroptosis [166].

Because mitochondrial destruction that is accompanied by destroyed cristae and a high electron density is characteristically observed in ferroptotic cells [9,167], ETC could be a primary source for radical electrons at least under some types of Cys deficiency. Inhibition of individual ETC, such as rotenone for ETC I and antimycin A for ETC III, has long been employed for studying respiratory chain function and oxidative stress associated with the mitochondrial respiratory chain. The application of these ETC inhibitors in cell culture studies reportedly mitigated ferroptosis under Cys–deprivation conditions, which implies that electrons that originated from mitochondrial ETC are responsible for lipid peroxidation and subsequent cell death [168]. However, these conventional inhibitors completely arrest the primary electron flow and may affect other mitochondrial metabolism. The prolonged inhibition of ETCs eventually causes cell damage or death due to energy consumption. In order to eliminate potential artifacts caused by the interrupted electron flow, it would be ideal to use inhibitors that specifically suppress electron leakage from the ETC. Brand et al. [169] recently developed some novel inhibitors, as exemplified by site I_Q_ electron leak (S1QEL) and suppressors of site III_Qo_ electron leak (S3QEL), which specifically block superoxide production from ETC I and ETC III, respectively, but do not affect the primary electron flow or normal ATP synthesis [168,170,171]. By employing S1QEL and S3QEL, we found that ETC III is the major source for production of superoxide responsible for ferroptosis under Cys–deprivation conditions [172].

Next to heme, iron-sulfur clusters such as 4Fe-4S and 2Fe-2S are an abundant iron-complex and a cofactor for electron-transfer proteins. Iron regulatory proteins (IRPs) are typical Fe-S cluster proteins that are present in two forms, namely, IRP1 and IRP2 [173]. While cytosolic IRP1 binds the iron-responsive element in mRNAs that are related to iron metabolism and regulates their translation, hence maintaining iron homeostasis, mitochondrial IRP2 acts as an aconitase in the TCA cycle. Superoxide that is released from mitochondrial ETC can target enzymes containing 4Fe-4S [11]. 4Fe-4S in aconitase is the most sensitive to superoxide and other ROS species [174,175]. The rapid reaction of superoxide with aconitase (k ~ 10^7^ M^−1^s^−1^) converts the 4Fe-4S cluster to the 3Fe-4S cluster and releases ferrous iron and hydrogen peroxide, which results in the formation of hydroxyl radicals via Fenton-type chemistry [176]. Moreover, the inactivation of aconitase may disrupt the mitochondrial TCA cycle and alter carbon metabolism [177]. Based on these collective bodies of evidence, we propose a possible pathway for ferroptosis in which radical electrons leaked from mitochondrial ETC play a role in the release of free iron from iron cluster and collectively cause lipid peroxidation reactions for executing ferroptosis (Figure 3). In this hypothetical mechanism, the resulting ferrous iron, together with hydrogen peroxide, causes the peroxidation of PUFA, which may consequently cause mitochondrial impairment. This would also explain the aberrant mitochondrial morphology observed in ferroptotic cells [9]. Succinate dehydrogenase is another enzyme that requires 4Fe-4S and is sensitive to superoxide in the TCA cycle. It would be intriguing to see if these Fe-S-containing enzymes are actually impaired under ferroptotic stimuli.

In addition to conventional Fe-S cluster proteins, there is a new class of 2Fe-2S-containing proteins, namely, the CDGSH iron sulfur domain 1 (CISD1) and the nutrient-deprivation autophagy factor-1 (NAF-1), also referred to as CISD2 [178]. These proteins are localized in the ER and/or the outer mitochondrial membrane surface and appear to be responsible for transferring [Fe-S] clusters. CISDs may be responsible for the 2Fe-2S cluster relay from mitochondria to other cellular components [179]. While CISD1 and CISD3 protect mitochondria in cells that have been treated with erastin and negatively regulate ferroptosis [180,181], CIDS2 appears to confer resistance against the ferroptosis caused by sulfasalazine, an inhibitor for xCT, in head and neck cancer cells [182]. We observed that CISDs suppress free iron toxicity under conditions where GSH synthesis is inhibited by buthionine sulfoximine treatment, which implies a role of Cys that is independent of GSH synthesis [116]. Fe-S clusters are synthesized in mitochondria by the action of the Fe-S cluster assembly protein ISCU, which regulates iron status and suppresses the ferroptosis induced by dihydroartemisinin [183]. Cysteine desulfurase enzyme NFS1 supplies Cys–derived sulfur for assembling Fe-S and reportedly exerts a protective action against ferroptosis in lung tumor cells [184]. Consistently, frataxin, which activates the cysteine desulfurase complex, negatively regulates ferroptosis by modulating the assembly of Fe-S clusters [185]. There appears to be a tight association between free iron and Fe-S clusters in ferroptosis in which mitochondria are deeply involved.

#### 4.2.3. Roles of Lipid Peroxidation in Ferroptosis

Figure 4 depicts a role of lipid damage initiated by arachidonate lipoxygenase (ALOX). Compounds such as vitamin E and ferrostatin-1 or enzymes such as GPX4 suppress lipid peroxidation and effectively rescue cells from ferroptosis [167,186]. The levels of lipid peroxidation products are elevated in cells typically by means of either the depletion of intracellular Cys or the inhibition of GPX4. Because Cys restricts the amounts of GSH synthesized in cells, the availability of Cys determines GSH synthesis and thereby the capacity of GPX4 for reducing phospholipid hydroperoxides (Figure 4). Elevated lipid peroxidation, together with free iron, plays an essential role in executing ferroptosis [187]. Given the involvement of free iron, it is conceivable that radical species, most likely hydroxyl radicals produced via Fenton chemistry, are responsible for the lipid peroxidation reaction [188]. Accordingly, either a suppressor of lipid peroxidation products or iron chelation effectively renders cells resistant to ferroptosis. This notion is clearly supported by the observation that edaravone, a radical scavenging compound, suppresses both lipid peroxidation and ferroptosis [189]. A redox protein, which is designated as the ferroptosis-suppressor protein 1 (FSP1), suppresses ferroptosis, and this action is independent from GPX4 activity, proceeding through the NAD(P)H-dependent reduction of ubiquinone (CoQ) after trapping lipid peroxyl radicals [190,191].

Because of their high nucleophilicity, hydroxyl radicals peroxidize PUFA, leading to the production of lipid hydroperoxides (LOOH) [192]. The resulting LOOH can be converted into alkoxyl radical, thus allowing the radical chain reaction to continue. An inactivation/inhibition of GPX4 activity causes the accumulation of phospholipid hydroperoxides, which would result in enhanced production of alkoxyl radicals in the presence of free iron and eventual destruction of the cell membrane [193]. Because acyl-CoA synthetase 4 (ACSL4), which catalyzes the formation of acyl-CoA preferentially using arachidonate, executes ferroptosis, the incorporation of peroxidized PUFA into phospholipids, notably phosphatidylethanolamine, may promote ferroptosis [194,195]. Protein kinase CβII (PKCβII) reportedly senses lipid peroxidation products and activates ACSL4 by phosphorylation at specific threonine, which results in the incorporation of PUFA into phospholipids and the enhancement of lipid peroxidation in a positive-feedback manner [196]. Despite extensive studies, however, the issue of how peroxidized phospholipids induce membrane destruction remains largely unknown [197].

Lipid peroxidation occurs by means of either enzymatic reactions that utilize oxygen molecules or non-enzymatic reactions that are mediated by ROS. Enzymatic oxidation by ALOX introduces oxygen into arachidonate and results in the production of lipid peroxides, which leads to eicosanoid production. Enzymes, such as ALOX12/ALOX15, that introduce oxygen into arachidonate in the initial step of eicosanoid formation have been identified as ferroptosis inducers under some types of stimuli [195,198]. When the resulting 15-hydroperoxyeicosatetraenoyl acid (15-HpETE) is incorporated into phospholipids by means of ACSL4 followed by lyso-phospholipid acyltransferase (LCAT) [188,194], it may trigger membrane destruction when free iron is present. Conversely, GPX4 interrupts the fatal chain reaction by reducing 15-HpETE to 15-hydroxyeicosatetraenoyl acid (15-HETE).

Irrespective of GPX4 activity, calcium-independent phospholipase A2β (iPLA2β) exerts anti-ferroptotic activity [199,200]. Because PUFA and peroxidized fatty acids are typically attached at the sn-2 position where iPLA2β acts on, the anti-ferroptotic action of iPLA2β may be attributable to direct removal of peroxidized PUFA from the phospholipid. As mentioned above, PRDX6 has bifunctional activities that include non-selenium phospholipid hydroperoxide GPX activity and iPLA2-like activity [141]. The iPLA2-like activity of PRDX6 is enhanced at acidic pH and in the presence of an oxidized phospholipid, such as oxidized sn-2 linoleic or arachidonic acid. Both of these activities could protect membrane phospholipids from oxidative damage and rescue cells. It has been reported that PRDX6 suppresses ferroptosis by decreasing the levels of lipid peroxidation products [201]. Because the phospholipid hydroperoxide GPX activity and iPLA2-like activity are located in structurally different domains of PRDX6 [141], it would be of interest to know which activity is actually involved in the anti-ferroptotic action of this compound. Given the fact that PRDX6 is overexpressed predominantly in breast cancer, lung adenocarcinoma, and melanoma [202], their malignant phenotype may be associated with the anti-ferroptotic action attributed to the intrinsic activities of PRDX6 toward peroxidized phospholipids.

#### 4.2.4. Other Factors Involved in the Regulation of Ferroptosis

The negative and positive regulators of ferroptosis are largely distinct from those for other types of cell death, which include apoptosis, necroptosis, pyroptosis, and parthanatos [203]. Although ferroptosis shares few genes responsible for other types of cell death, the activated p53 tumor-suppressor protein effectively induces ferroptosis via a transcription-dependent or post-translation-dependent mechanism [204]. The first report on this issue showed that an acetylation-defective mutant p53 failed to induce cell cycle arrest, senescence and apoptosis, but regulated the expression of xCT and induced ferroptosis [205]. Subsequent studies have revealed the existence of multiple actions of p53 resulting in the stimulation or suppression of ferroptosis, depending on the types of cells and the conditions being employed [206]. For example, dipeptidyl peptide 4 (DPP4) in the plasma membrane activates NOX1 to produce ROS and stimulate ferroptosis [207]. p53 binds DPP4 and is translocated to the nucleus where the complex transcriptionally activates the expression of SLC7A11, the gene encoding xCT, resulting in the suppression of ferroptosis.

Enzymes that consume oxygen molecules are surely candidates as radical sources for ferroptosis. In addition to ETC, pyruvate dehydrogenase (PDH), 2-oxoglutarate dehydrogenase (OGDH), and branched-chain 2-oxoacid dehydrogenase (BCKDH) complexes are capable of producing superoxide/hydrogen peroxide, and a higher production of superoxide is observed in OGDH and PDH complexes than for ETC I [13]. OGDH activity is increased upon ferroptotic stimuli, i.e., Cys deprivation and glutaminolysis, in certain types of cancer cells. Circumstantial evidence implies that the reaction of dihydrolipoamide dehydrogenase in the PDH complex, which links glycolysis and the TCA cycle, is also associated with ferrroptosis execution [208]. Dihydrolipoamide produced during the catalytic action of the PDH complex appears to become the source of superoxide [209]. Moreover, POR, which provides electrons for the CYP reaction, is also responsible for the execution of ferroptosis via the generation of hydrogen peroxide by transferring electrons to oxygen [210]. Consistently, POR knockdown in mice prevents concanavalin A-induced acute liver hepatitis in which ferroptosis appears to play a role [211]. Because there are many enzymes that consume oxygen molecules and potentially produce ROS, the number of enzymes responsible for the execution of ferroptosis will likely increase.

## 5. Antioxidative Metabolites to Combat Ferroptosis

GPX requires GSH for its catalytic properties, which explains why a Cys/GSH deficiency results in ferroptosis through GPX4 incompetence. Due to its significant and pleiotropic roles, GSH synthesis and function have been extensively investigated for over a century, e.g., [113]. Multiple micronutrients and metabolites with low molecular sizes have been reported to exert antioxidative activity and exert compensatory roles for antioxidative enzymes. These antioxidative compounds react stoichiometrically with ROS, and if not properly recycled or replenished, they become depleted. Toc and Asc are micronutrients that are fairly abundant in the body and play pivotal roles in antioxidation. In addition, some metabolites also act in the antioxidation. NO, as an example, is generally regarded as a cellular signaling molecule but also vigorously reacts with radicals and other ROS and, hence, can be regarded as an antioxidant from another point of view. These antioxidative properties of micronutrients and the metabolites appear to be either directly or indirectly involved in the suppression of ferroptosis through abstracting or neutralizing the causative radical species, as shown in the scheme in Figure 5.

### 5.1. Vitamin E Suppresses Lipid Peroxidation

Vitamin E is a natural lipophilic radical scavenger and consists of three compounds α-, β-, and γ-tocopherol that have to be ingested. The donation of one electron from an alkoxyl radical to Toc results in the production of a tocopheroxyl radical and lipid peroxide [192]. While tocopheroxyl radicals are less reactive and can be recycled via donating the radical electron to other compounds such as Asc, the resulting lipid peroxides are reduced to the corresponding alcohols by the action of peroxidases, notably GPX4. Thus, the coordinate action of Toc, Asc, GPX4 can suppress ferroptosis [188]. Because Toc is extremely hydrophobic and is mainly localized in membranes or lipid droplets, several compounds, trolox for example, have been synthesized in attempts to overcome the low solubility.

### 5.2. Vitamin C Is a Hydrophilic Antioxidant but Suppresses Lipid Peroxidation

Asc (vitamin C in primates), which is present the submillimolar range within cells, plays multiple roles not only by supporting a variety of enzymatic reactions in which redox reactions are involved but also in antioxidation [212]. As an important antioxidant, Asc has also been overviewed from the standpoint of chemical and biological reactions [213,214]. However, the interaction of Asc with iron also causes a potentially harmful situation by stimulating ROS production [215]. Thus, the issue of whether Asc acts as a prooxidant or an antioxidant greatly depends on the physiological conditions and the microenvironment of the body.

While plants produce Asc-dependent peroxidases, higher animals do not. Hence, the antioxidant effects of Asc are solely mediated by non-enzymatic reactions in mammals. Asc reacts directly with superoxide, which results in the formation of an Asc radical and hydrogen peroxide. The resulting Asc radical is either reduced back to Asc or further oxidized by the action of another radical species. The second-order rate constant for the reaction between Asc and superoxide is 1 × 10^5^ M^−1^s^−1^, while that for the Asc radical and superoxide is 2.6 × 10^8^ M^−1^s^−1^ [214]. The Asc radical also reacts preferentially with other radicals that include tocopheroxyl radicals (k = 7 × 10^7^ M^−1^s^−1^) and peroxyl radicals (k ~ 2.0 × 10^6^ M^−1^s^−1^) [213,216]. Accordingly, the presence of abundant levels inside cells allows Asc to act as a scavenger of superoxide and other radical species. The interaction of superoxide and Asc may be the cause for the low Asc levels in SOD1 and SOD3 double-deficient mice [101]. Aldehyde reductase (Akr1a) is involved in Asc synthesis through the catalytic conversion of D-glucuronic acid to L-gulonate in the Asc synthesis pathway, such that Akr1a-deficient mice produce approximately 10% Asc of wild-type mice [217]. The double knockout of Akr1a and SOD1, therefore, has pathological consequences due to increased superoxide, i.e., death within two weeks, irrespective of their sex or ages [117]. The lung is the most affected organ in the Akr1a- and SOD1-double knockout mice, which is consistent with the presence of the highest concentrations of oxygen in body. A fatal phenotype has been also reported in mice with a double deficiency of SOD1 and gluconolactonase, which catalyzes the penultimate reaction in the synthesis of Asc [218], although the cause for fatality of the mice has not been extensively examined from the aspect of oxidative insult. Nevertheless, the superoxide-scavenging action of Asc appears to be essential for the survival of mice under conditions of an excessive presence of superoxide due to SOD1 deficiency.

### 5.3. Nitric Oxide as Both Pro-Oxidant and Antioxidant

Nitric oxide (NO) has been identified as the vascular endothelium-derived vasodilation factor in biological systems [219]. After the finding of NO production in the body, nitric oxide synthase (NOS) was isolated and characterized as an intrinsic source of NO. NO is produced via the six-electron reduction of nitrogen in the guanidino group of arginine. Three isoforms, namely, the neuronal form (NOS1 or nNOS), the inducible form (NOS2 or iNOS), and the endothelial form (NOS3 or eNOS) have been identified in the mammalian body [220]. Processes in which NOS catalyzes the conversion of arginine and molecular oxygen to citrulline and NO have been extensively investigated [7]. We now know that NO is also produced by other enzymatic or non-enzymatic reactions [6]. Although it has been revealed that NO is involved in numerous actions, we focus here on radical scavenging actions from the perspective of antioxidants.

#### 5.3.1. NO as a Potent Superoxide Scavenger

While gene ablation of NOS isozymes shows some abnormalities in mouse phenotypes, significant amounts of NO are still produced in an NOS-independent manner and appear to complement the absence of NOS, thereby mitigating the phenotypic abnormalities in the NOS-deficient mice. For example, nitrite–nitrate is the source for NO by chemical reactions in the stomach under acidic conditions [221]. Hemoglobin (Hb) in red blood cells catalyzes the reductive conversion of nitrite to NO via a series of complex chemical reactions [222,223]. As a result, Hb is oxidized to methemoglobin but can be reduced back by methemoglobin reductase with NAD(P)H as the electron source [224]. The best described NO-synthesizing enzyme in plants is a molybdenum-dependent enzyme, nitrate reductase, but their homologue is not present in animals. Conversely, a homologue to mammalian NOS is not present in plants. Instead, molybdenum-dependent enzymes consisting of XOR, sulfate oxidase, mitochondrial amidoxime reducing component, and aldehyde oxidase are all reportedly able to reduce nitrite to NO in mammals [6]. For example, molybdenum in XOR plays a primary role in the reduction of nitrite to NO [225,226]. The physiological significance of NO produced from sources other than NOS becomes evident in mice that have been fed a diet deficient in nitrite–nitrate for long periods of time [227]. These mice possess NOS isozymes but develop metabolic syndrome and endothelial dysfunction and eventually die from cardiovascular dysfunction, suggesting essential roles of NO derived from sources other than NOS.

Because of its radical nature, the reaction of superoxide and NO proceeds in a diffusion-limited manner (k = 1.9 × 10^10^ M^−1^s^−1^) [228], which is faster than the SOD-catalyzed dismutation of superoxide. The benefit of scavenging superoxide is that it eliminates radical electrons at an early stage and hence shuts down subsequent radical chain reactions [1]. Given the abundant production of NO, the superoxide scavenging ability by NO appears to be comparable to SOD activity inside cells, and it could be much more efficient than that in an extracellular environment.

Resulted ONOO^−^ is a strong oxidant and is toxic to cells when produced abundantly beyond antioxidative capacity [8]. However, there is emerging evidence to suggest a beneficial action of ONOO^−^. In the case of the vascular system, for example, moderate levels of ONOO^−^ appear to stimulate prostanoid synthesis and act in cellular signal-transducing reactions [229]. The advantage of NO in scavenging superoxide is supported by the observation that higher levels of superoxide and hydrogen peroxide are present in alveolar macrophages from NOS2-knockout mice compared to those from wild-type mice [230]. In vitro studies indicate that the oxidation of low-density lipoprotein (LDL) is more evident by incubation with NOS2-knockout macrophages than WT macrophages, upon stimulation with interferon-γ [231]. The elevation in LDL oxidation is consistently suppressed by an NO donor, suggesting that preserved superoxide due to the absence of NOS2 rather than the resulting ONOO^−^ stimulates LDL oxidation. The beneficial action of NO produced under inflammatory conditions has been also reported in pathological model animals [232], although some reports show the opposite. Our recent study using primary macrophages isolated from mice lacking NOS2 and SOD1, either singly or doubly, clearly demonstrate a protective action of NO against superoxide toxicity under SOD1 deficient conditions [233]. The viability of macrophages from SOD1-knockout mice is markedly ameliorated by both an exogenous NO donor and endogenously produced NO, despite an increase in the production of ONOO^−^ under such conditions.

The cytotoxic effects of ONOO^−^ may be mitigated by the actions of scavenging molecules in vivo. Some enzymes that include GPX1 [234] and PRDX2 [235] reportedly function as peroxynitrite reductases. The rate constant for the ONOO^−^ decomposition by human PRDX2 is 1.4 × 10^7^ M^−1^ s^−1^ at 25 °C and pH 7.4 [236]. Because inflammatory cells simultaneously produce both superoxide and NO, these observations suggest that the NO-mediated detoxification of superoxide overcomes cytotoxic action of ONOO^−^ under such circumstances.

#### 5.3.2. NO in Termination of Radical Chain Reactions in Lipid Peroxidation

GPX4 is the primary anti-ferroptotic enzyme that reductively removes LOOH, thereby suppressing the production of alkoxyl radicals (LOO^.^), which appears to cause destruction of the membrane [188,237]. The reaction of NO with LOO^.^ forms stable nitrogen-containing products, LONO and LOONO, and terminates the chain reaction of lipid radials [238]. NO reacts with LOO^.^ in a nearly diffusion-limited reaction (k = 1–3 × 10^9^ M^−1^s^−1^) [239], which is two to three orders of magnitude higher than that for the reaction between LOO^.^ and Toc, a physiological anti-ferroptotic compound [240]. Thus, NO, if produced in sufficient amounts, is a potent scavenger for LOO^.^ and, hence, may act as a physiological suppressor for apoptosis, as observed in endothelial cells that are exposed to oxidized LDL [241]. Accordingly, the antioxidant effect of NO on lipid peroxidation may, at least in part, be explained by this termination of the radical chain reaction [242]. A recent study revealed that, under pro-inflammatory conditions, macrophages that produce a large body of NO suppress ferroptosis via the nitroxygenation of 15-hydroperoxy-eicosa-tetra-enoyl-phosphatidylethanolamine (15-HpETE-PE) [243]. We found that a long-lasting NO donor effectively suppressed the ferroptosis that is induced in mouse hepatoma-derived Hepa1-6 cells under Cys deprivation cultures, xCT inhibition, and GPX4 inhibition [244]. Although we did not identify LONO/LOONO in the cells, circumstantial evidence suggests that the reaction of NO with LO/LOO^.^ is responsible for aborting ferroptosis via suppressing the radical chain reaction. Given the involvement of radicals in ferroptosis induced by lipid peroxidation reactions, other compounds that interact with radical species may also act more or less as suppressors of ferroptosis.

## 6. Concluding Remarks

Superoxide is primarily produced via metabolic reactions in which oxygen molecules are consumed. Because the production of oxygen radicals is enhanced under conditions of stimulated metabolism with oxygen consumption, ferroptosis likely occurs in metabolically active cells. While Toc and Asc exert antioxidative actions via the mitigation of toxicity of radical electrons in superoxide and other radical species, NO binds superoxide and alkoxyl radicals and terminates their radical electrons. Accordingly, SOD as well as these antioxidant compounds appear to suppress ferroptosis by inhibiting radical chain reactions, either directly or indirectly.

## Figures and Tables

**Figure 1 antioxidants-11-00501-f001:**
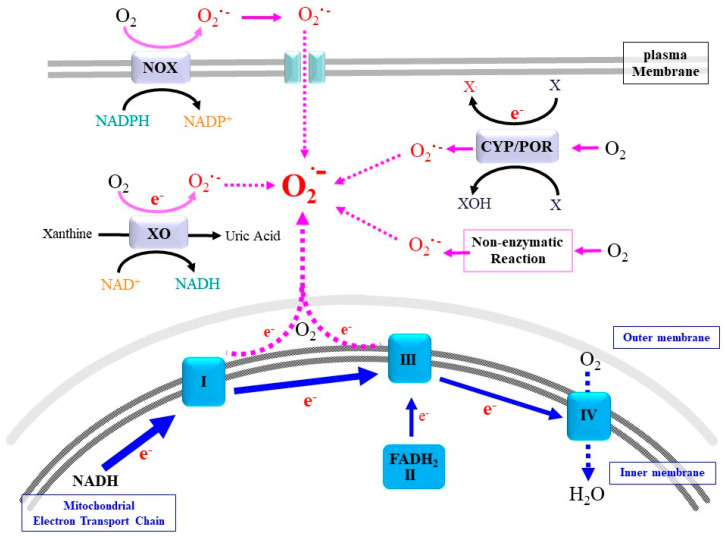
Representative electron sources of superoxide. While mitochondrial ETC is the major source for superoxide (O_2_^.−^), many enzymes, such as NADPH oxidase (NOX), xanthine oxidase (XO), and cytochrome P_450_ (CYP)/cytochrome P_450_ reductase (POR), convert molecular oxygen to superoxide either as a main product or as a byproduct during oxidation of a variety of compounds (X), such as benzene compounds, drugs, and steroid hormones. Superoxide is also produced non-enzymatically.

**Figure 2 antioxidants-11-00501-f002:**
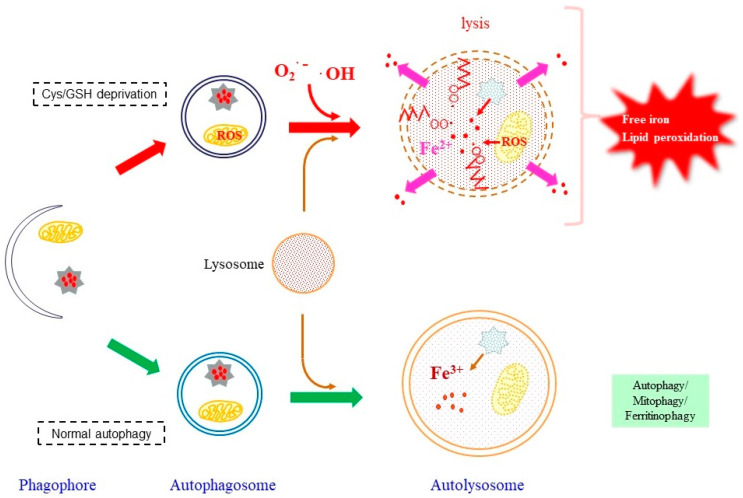
A role of ferritinophagy in releasing free iron. In normal ferritinophagy, free iron is released from ferritin but remains in the autolysosomes, which does not cause deteriorating effects as schematized at the bottom. Under ferroptotic conditions, the produced ROS causes lipid peroxidation. The presence of peroxidized lipids and free iron would make autolysosome membrane vulnerable, leading to the destruction of the vesicular structure.

**Figure 3 antioxidants-11-00501-f003:**
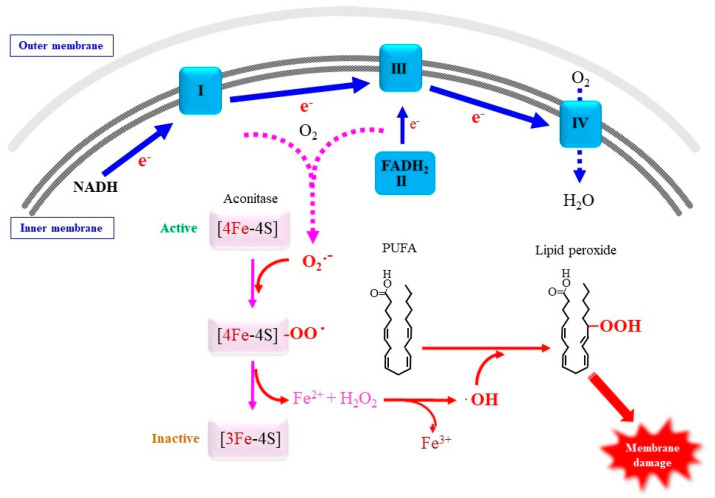
Proposed role of superoxide in mitochondrial membrane damage. Superoxide that is released from the ETC under ferroptotic stimuli may become attached to 4Fe-4S clusters in proteins, notably aconitase, which results in the release of ferrous iron and hydrogen peroxide. PUFA in mitochondrial membrane undergoes peroxidation by ROS and consequently destructs mitochondrial membrane integrity.

**Figure 4 antioxidants-11-00501-f004:**
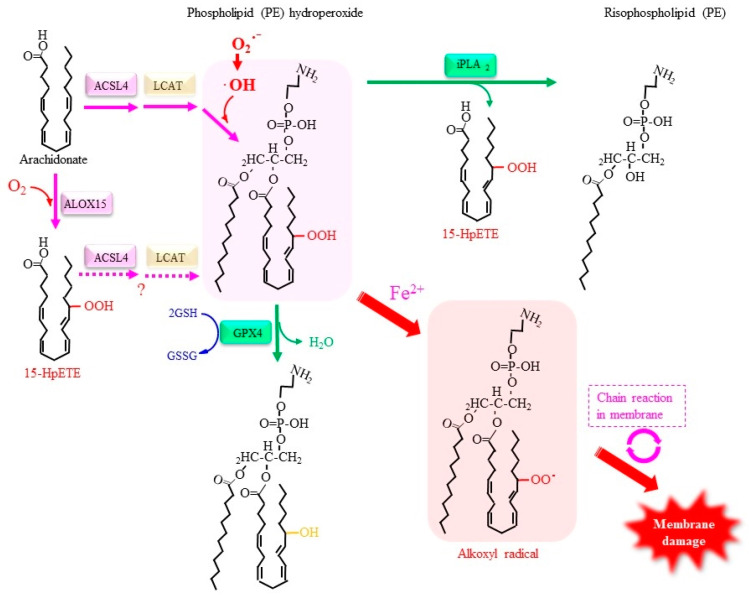
ALOX15 is a representative enzyme that catalyzes the peroxidation of arachidonate. ACSL4 preferentially acts on arachidonate derivatives and forms acyl-coenzyme A, which is then used to construct phospholipids by LCAT. Lipid peroxides in phospholipid bilayers react with free iron and result in the production of alkoxyl radicals, which enhances lipid peroxidation, ultimately leading to the destruction of the membrane structure. iPLA2 preferentially removes an acyl group at the sn-2 position where an unsaturated fatty acid is conjugated such that 15-HpETE can be excised by the action of iPLA2, which leads to the membrane structure being protected. Conversely, GPX4 interrupts the fatal chain reaction by reducing 15-HpETE to 15-HETE in the form of phospholipid hydroperoxide.

**Figure 5 antioxidants-11-00501-f005:**
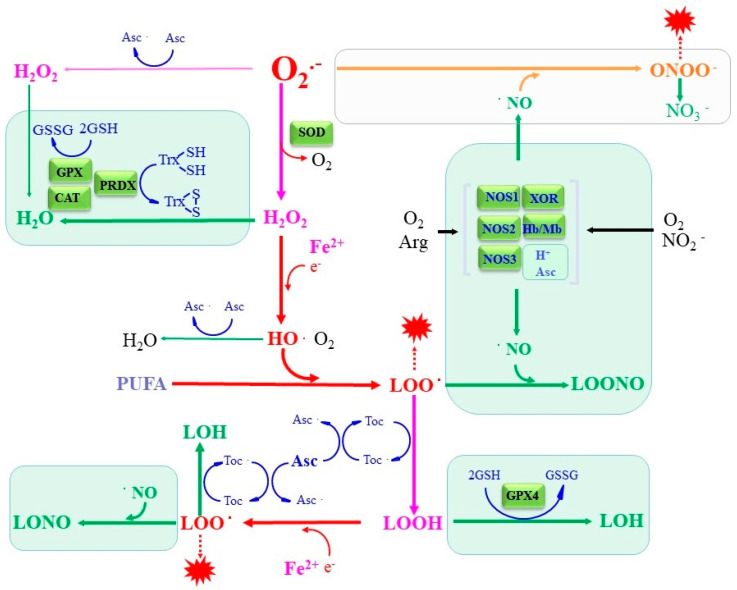
The interaction between ROS/lipid radicals with antioxidants in vivo. Detoxification reactions performed by representative antioxidative enzymes and low molecular weight antioxidants are presented in green. In addition to antioxidative enzymes, Toc, Asc, and NO can terminate radical chain reactions, which result in the suppression of ferroptosis.

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
