# Peer review of "Superoxide Radicals in the Execution of Cell Death"

_antioxidants, 2022, doi:10.3390/antiox11030501_

Round 1
Reviewer 1 Report
The review by Junichi Fujii et al. is very interesting and timely. It is well written and cover most of the biological processes linked to the ROS. However, I think there are some areas that still needs to be discussed in this review such as
- Role of superoxide ions in the stress granule formation in the cell
- Influence of these stresses in different neurodegenerative diseases, such as Alzheimer’s and Parkinson’s disease.
Author Response
We appreciate the reviewers' evaluation of the manuscript and their advice. We have amended it mostly according to your comments and advice.
Responses:
We mentioned Down syndrome and ALS because they are typical diseases that aer directly associated with SOD1 gene disorders. We understand how important the issues raised by the reviewer are. In fact, we could not address many important issues on superoxide, such as pulmonary diseases, cardiovascular diseases, infertility, etc... Unfortunately, however, it has been lengthened to spend space for the description of cell death, so that we wish to discuss them at next chance.
Reviewer 2 Report
This review provides a comprehensive and very well-crafted overview about the role of superoxide in the execution of cell death.
Detailed Comments
Line 10: It should read ‘role in the cellular defense’.
Line 14: It should read ‘has attracted considerable’.
Line 15: ‘Radical electrons, namely mitochondrial electron transfer complexes’ does not make sense and should be reformulated.
Line 19: It should read ‘that includes’.
Line 20+21: ‘neutralize radical electrons in corresponding target compounds’ is unclear and must be reformulated. Is it possible to just remove ‘in corresponding target compounds’?
Line 38: It is suggested to reformulate the sentence to ‘Excessive intracellular ROS concentrations tend to result in the oxidative modification of vital biomolecules which can lead to accelerated ageing and the aggravation of certain disease processes.’.
Line 49: It should read ‘from another point of view’.
Line 55: It should read ‘enzymes that maintain’.
Line 56: It should read ‘adequate levels.’.
Line 58: It should read ‘discuss the significance’.
Line 60: It should read ‘(Asc) to maintain (i.e. remove the comma).
Line 68: ‘The followings’ is not a good formulation. It could be replaced by ‘In the next paragraphs we provide an overview on the major’.
Line 70: It should read ‘damage to vital cellular biomolecules.’.
Line 73: The abbreviations ’NOX, XO and CYP/POR’ must be provided in the figure caption to enhance clarity. In addition, it should be clarified what ‘X’ refers to in the top right of the figure and an example should be provided which helps the reader to understand what ‘X’ may be (an environmental pollutant?) to enhance clarity.
Line 88: It should read ‘The ETC eventually’.
Line 90: It should read ‘superoxide anions are produced.’.
Line 97: It should read ‘TCA cycle can also serve’.
Line 105: It should read ‘The produced ROS can then be released to the opposite side of the cytoplasm’.
Line 106: It should tread ‘where they can modulate’.
Line 107: It should read ‘however, can oxidatively modify’.
Line 109: It should read ‘family of proteins’.
Line 116: It should read ‘and releases superoxide.’.
Line 117: To the best of my knowledge I have never read that ‘perchloric acid’ is formed within cells. Please double check if this species represent an ROS species.
Line 128: it should read ‘mechanism operates in many’.
Line 129: ‘including receptors for hormones and immune systems’ is unclear and must be reformulated.
Line 136: It should read ‘PTPs are transiently inactivated during an oxidative insult’.
Line 137: ‘being sustained’ is unclear. Could it be removed to enhance the clarity of this sentence?
Line 142: It should read ‘and affect signal transduction’?
Line 145+6: ‘is in a low pKa state’ is unclear. Please reformulate.
Line 150: It should read ‘by other reducing biomolecules’.
Line 151: ‘have unrestricted targets’ is not a good word choice. Would ‘ROS indiscriminately target biomolecules, they’.
Line 157: It should read ‘Cys-sulfonic acid’ (i.e. remove the ‘r’).
Line 158: It should read ‘hyperoxidized Cys species back to’.
Line 164: ’PTEN’ is unclear and must be explained (what does this abbreviation refer to?).
Line 170: ‘make the ROS signal appropriate’ is not a good formulation and could be improved. ‘to maintain ROS signaling in an appropriate range’.
Line 188: After ‘anti-cancer drugs.’ an appropriate reference must be provided.
Line 189: It should read ‘is still unclear.’.
Line 193: ‘with the oxidase reaction’ is unclear. Please reformulate what is meant here.
Line 202: It should read ‘for the detoxification’. Thereafter, please add ‘what’ is being detoxified.
Line 203: It should read ‘may be associated with’.
Line 213: ‘and reserved electrons may be utilized’ is unclear and needs to be reformulated.
Line 215: ‘or limited proteolysis’ is unclear (proteolysis of what?). Please reformulate.
Line 221: After ‘nitrite reduction’ please add information as to where this unfolds (in the liver?).
Line 226: After ‘treatment’ can information be provided for which disease specifically.
Line 234: It should read ‘hyperglycaemic conditions’.
Lien 238: It should read ‘compounds to protect against’.
Line 252: After ‘broad pH range’ the pH range should be provided [e.g. (pH 3-8)].
Line 253: ‘electrostatic interactions’ of what in the active center. Please clarify/reformulate.
Line 257: It should read ‘than today [50].’.
Line 258: can some detail be provided about the organs in which these (SOD1-3) are found.
Line 259: It should read ‘we here outline’.
Line 260: ‘aspects of controlling the activity’ needs to be reformulated.
Line 264: It would bee useful if it could be specified in which cytosol this SOD is present in humans (liver, kidneys, brain, etc.).
Line 272: It should read presence of a CCS-independent’.
Line 275: It should read ‘from the Cu status of the organism [56].’.
Line 279: Should it read ‘with an altered SOD1 activity’?
Line 289: It should read ‘SOD1 consumes’.
Lime 296: It should read ‘Several mouse models have been established’.
Line 308: It should read ‘SOD1 activity are attributable’, ‘to the transgene’ is unclear and should be reformulated.
Line 314: It should read ‘Collectively, this evidence suggests that’.
Line 315: It should read ‘cause for the emergence’.
Line 328: It should read ‘antioxidative properties’.
Line 331: It should read ‘caloric restriction’.
Line 345: ‘toxins’ is unclear. Some detail must be provided as to what compounds are meant here.
Line 353: It should read ‘vascular dysfunction compared to control mice [85].’.
Line 368: ‘maintenance of lung homeostasis’ could be better formulated.
Line 383: It should rad ‘In an initial study’.
Line 386: ‘potential roles of the gene; should be reformulated.
Line 403: Instead of ‘to permissive levels’ could be replaced by ‘to manageable levels’.
Line 406: It should read ‘in peroxisomes of’.
Line 433: Given the important of Cys and GSH in the remainder of the manuscript it would be helpful to the reader if the range of free Cys and GSH in mammalian cells (e.g. hepatocytes) could be provided.
Line 437: It should read ‘, but does not cause a decrease in’.
Line 438: ‘when used for a short time’ is unclear. Do the authors mean as a dietary supplement?
Line 445: ‘other compounds’ is unclear and more detail about these must be provided.
Line 462: It would be helpful to the reader if the conc. range of Glu in the cell cytosol could be provided.
Line 476: Remove the comma after GPX8.
Line 482: ‘in such deaths’ is not a good formulation. “in this instance’ may be a better formulation.
Line 490: It should read ‘GPX family is provided in a recent review article [130]’.
Line 495: the abbreviation ‘PRDX’ must be explained.
Line 500: ‘including chaperone activity’ please explain for which metal in particular?
Line 503: ‘in another partner’ is unclear and must be reformulated.
Line 532: It should read ‘Other mechanisms of non-apoptotic cell death are classified’.
Line 546: It should read ‘in the literature’.
Line 552: ‘are self-divided’ is not a good formulation. Could it read ‘cells disintegrate into small’.
Line 559: Remove ‘which’ and it should read ‘also appear to largely’.
Line 564+5: It should read ‘examined in great detail.’ And ‘phospholipid that interacts’.
Line 574: ‘performs catalysis’ please add of what?
Line 578: ‘L/S’ is unclear and needs to be explained. The sentence from his line to line 580 is unclear and needs to be reformulated.
Line 619: It should read ‘under Cys deficiency [156].’. To improve clarity it would be great if the cytosolic conc. of free Cys could be provided here.
Line 621: It should read ‘deficiency’. Can details be provided with regard to what constitutes Cys deficiency (cytosolic Cys conc. < 0.05 mM)?
Line 623: It should read ‘represented’.
Line 629: It should read ‘of Cys deficiency’.
Line 632: It should read ‘lysosome is the organelle’.
Line 633: Should it read ‘than other subcellular compartments’ instead?
Line 636: It should read ‘ferrous’.
Line 637: It should read ‘out of the autolysosome’.
Line 638: It should read ‘fragile which may potentially result in its rupture’.
Lien 640: It should read ‘the integrity of the cell membrane’.
Line 656: It should read “Proposed role of’.
Line 664: It should read ‘have been recently reported’.
Line 668: It should read ‘which initiate ferroptosis’.
Line 671: It should read ‘Cys deficiency’.
Line 675: It should read ‘studies reportedly mitigated ferroptosis’.
Line 702: It should read ‘play a role in the’.
Line 715: After ‘clusters’ the authors need to clarify from where to where these cluster proteins are transported.
Line 719: Please provide details (structure) of sulfasalazine for context.
Line 730: It should read ‘Figure 3 depicts a role of lipid’ and the abbreviation ‘ALOX’ must be explained.
Line 748: Detail of ferrostatin1 need to be provided for context.
Line 761: Please remove comma after (FSP1).
Lien 764; It should read ‘peroxidize PUFA’.
Line 766+7: It should read ‘An inactivation/inhibition of GPX4’.
Line 777: It should read ‘induce membrane destruction’.
Line 807: It should read ‘in the regulation of’.
Lien 810: ‘barely share responsible genes’ should be reformulated.
Line 811: It should read ‘effectively induces’.
Line 816: It should read ‘p53 resulting in the stimulation’.
Lien 865: ‘that are naturally present in our bodies’ should be changed to denote that they have to be ingested (one of the tocopherols is vitamin E, which needs to be orally ingested).
Line 873: ‘to overcome this issue’ it is unclear which issue the authors refer to. Please specify.
Line 883: It should read ‘of the body.’.
Line 900: It should read ‘in double’.
Line 901: Please specify if you mean SOD1 and SOD3 specifically.
Line 920: ‘from the standpoint of anti-oxidation’ is not a good formulation. Please reformulate.
Line 950: ‘ONOO- is toxic to cells’ is not a good formulation as the dose makes the poison. Please provide context (i.e. is toxic at rather low doses of xyz).
Line 961: Please remove the underlining of LDL.
Line 1005: ‘with active metabolism’ is not a good formulation.
Line 1010+11: Please remove.
Author Response
We appreciate the reviewers' evaluation of the manuscript and their advice. We have amended it mostly according to your comments and advice.
Line 10: It should read ‘role in the cellular defense’
Response: We have corrected it.
Line 14: It should read ‘has attracted considerable’.
Response: We have corrected it.
Line 15: ‘Radical electrons, namely mitochondrial electron transfer complexes’ does not make sense and should be reformulated.
Response: We have rephrased the sentence to “radical electrons, namely those released from mitochondrial electron transfer complexes, ...
Line 19: It should read ‘that includes’.
Response: We have corrected it.
Line 20+21: ‘neutralize radical electrons in corresponding target compounds’ is unclear and must be reformulated. Is it possible to just remove ‘in corresponding target compounds’?
Response: We have removed ‘in corresponding target compounds’ according to the advice.
Line 38: It is suggested to reformulate the sentence to ‘Excessive intracellular ROS concentrations tend to result in the oxidative modification of vital biomolecules which can lead to accelerated ageing and the aggravation of certain disease processes.’.
Response: We have rephrased it according to the suggestion.
Line 49: It should read ‘from another point of view’.
Response: We have corrected it.
Line 55: It should read ‘enzymes that maintain’.
Response: We have corrected it.
Line 56: It should read ‘adequate levels.’.
Response: We have corrected it.
Line 58: It should read ‘discuss the significance’.
Response: We have corrected it.
Line 60: It should read ‘(Asc) to maintain (i.e. remove the comma).
Response: We have corrected it.
Line 68: ‘The followings’ is not a good formulation. It could be replaced by ‘In the next paragraphs we provide an overview on the major’.
Response: We have replaced the statement with the advice one.
Line 70: It should read ‘damage to vital cellular biomolecules.’.
Response: We have added the word.
Line 73: The abbreviations ’NOX, XO and CYP/POR’ must be provided in the figure caption to enhance clarity. In addition, it should be clarified what ‘X’ refers to in the top right of the figure and an example should be provided which helps the reader to understand what ‘X’ may be (an environmental pollutant?) to enhance clarity.
Response: We have added full names of them for clarity and provided examples of X.
Line 88: It should read ‘The ETC eventually’.
Response: We have added “The”.
Line 90: It should read ‘superoxide anions are produced.’.
Response: We have corrected it.
Line 97: It should read ‘TCA cycle can also serve’.
Response: We have removed “the”.
Line 105: It should read ‘The produced ROS can then be released to the opposite side of the cytoplasm’.
Response: We have corrected the sentence according to the advice.
Line 106: It should tread ‘where they can modulate’.
Response: We have added “can” .
Line 107: It should read ‘however, can oxidatively modify’.
Response: We have added “can”.
Line 109: It should read ‘family of proteins’.
Response: We have added “of”.
Line 116: It should read ‘and releases superoxide.’.
Response: We have corrected it.
Line 117: To the best of my knowledge I have never read that ‘perchloric acid’ is formed within cells. Please double check if this species represent an ROS species.
Response: Thank you very much for pointing out. We have corrected ‘perchloric acid’ to “hypochlorous acid” and rephrased the sentence.
Line 128: it should read ‘mechanism operates in many’.
Response: We have corrected it.
Line 129: ‘including receptors for hormones and immune systems’ is unclear and must be reformulated.
Response: We have reformulated the sentence.
Line 136: It should read ‘PTPs are transiently inactivated during an oxidative insult’.
Response: We have corrected it.
Line 137: ‘being sustained’ is unclear. Could it be removed to enhance the clarity of this sentence?
Response: Thank you very much for pointing out. We have rephrased it to “, which results in sustaining the phosphorylation of signaling molecules.” for clarity.
Line 142: It should read ‘and affect signal transduction’?
Response: Thank you very much for pointing out. We have replaced it with ‘and affect signal transduction’.
Line 145+6: ‘is in a low pKa state’ is unclear. Please reformulate.
Response: We have corrected it to “…has a low pKa due to the structural microenvironment and…“.
Line 150: It should read ‘by other reducing biomolecules’.
Response: We have corrected it.
Line 151: ‘have unrestricted targets’ is not a good word choice. Would ‘ROS indiscriminately target biomolecules, they’.
Response: Thank you very much for pointing out. We have replaced the part with ‘ROS indiscriminately target biomolecules’.
Line 157: It should read ‘Cys-sulfonic acid’ (i.e. remove the ‘r’).
Response: We have corrected it.
Line 158: It should read ‘hyperoxidized Cys species back to’.
Response: We have added ”species”.
Line 164: ’PTEN’ is unclear and must be explained (what does this abbreviation refer to?).
Response: It is unusually long but “tensin homolog deleted from chromosome 10”is the full name for PTEN. We have corrected the full name and rephrased the sentence.
Line 170: ‘make the ROS signal appropriate’ is not a good formulation and could be improved. ‘to maintain ROS signaling in an appropriate range’.
Response: We have replaced the words with ‘to maintain ROS signaling in an appropriate range’ according to the advice.
Line 188: After ‘anti-cancer drugs.’ an appropriate reference must be provided.
Response: We have provided reference [3].
Line 189: It should read ‘is still unclear.’.
Response: We have corrected it.
Line 193: ‘with the oxidase reaction’ is unclear. Please reformulate what is meant here.
Response: We have rephrased the part to “.. releases superoxide as a byproduct of the oxidase reaction.”
Line 202: It should read ‘for the detoxification’. Thereafter, please add ‘what’ is being detoxified.
Response: Thank you for advice. The organ detoxifies “drug or xenobiotics” mentioned later. We have rephrased the sentence.
Line 203: It should read ‘may be associated with’.
Response: We have corrected it.
Line 213: ‘and reserved electrons may be utilized’ is unclear and needs to be reformulated.
Response: We have reformulated the sentence.
Line 215: ‘or limited proteolysis’ is unclear (proteolysis of what?). Please reformulate.
Response: We have reformulated the sentence. cysteine residues in or limited proteolysis of
Line 221: After ‘nitrite reduction’ please add information as to where this unfolds (in the liver?).
Response: We have added information “dominantly in cardiovascular system” after ‘nitrite reduction’
Line 226: After ‘treatment’ can information be provided for which disease specifically.
Response: We have added “of gout” after ‘treatment’.
Line 234: It should read ‘hyperglycaemic conditions’.
Response: We have corrected it.
Lien 238: It should read ‘compounds to protect against’.
Response: We have corrected it.
Line 252: After ‘broad pH range’ the pH range should be provided [e.g. (pH 3-8)].
Response: We have added the pH range on a specific enzyme “pH 5.3-9.5 in the bovine enzyme”.
Line 253: ‘electrostatic interactions’ of what in the active center. Please clarify/reformulate.
Response: The enzyme “SOD” comes later in this sentence, but we have reformulated the part.
Line 257: It should read ‘than today [50].’
Response: We have corrected it.
.
Line 258: can some detail be provided about the organs in which these (SOD1-3) are found.
Response: We did not provide the information on the organs because it appeared to be too detailed. However, we have added them.
Line 259: It should read ‘we here outline’.
Response: We have added ‘here’.
Line 260: ‘aspects of controlling the activity’ needs to be reformulated.
Response: We have corrected it.
Line 264: It would bee useful if it could be specified in which cytosol this SOD is present in humans (liver, kidneys, brain, etc.).
Response: We have corrected it “from aspects of regulation of their activities” .
Line 272: It should read ‘presence of a CCS-independent’.
Response: We have corrected it.
Line 275: It should read ‘from the Cu status of the organism [56].’.
Response: We have corrected it
Line 279: Should it read ‘with an altered SOD1 activity’?
Response: We have corrected it.
Line 289: It should read ‘SOD1 consumes’.
Response: Because the description is applicable for all SOD family enzymes, we have corrected it as “SOD family enzymes consume”.
Lime 296: It should read ‘Several mouse models have been established’.
Response: We have corrected it.
Line 308: It should read ‘SOD1 activity are attributable’, ‘to the transgene’ is unclear and should be reformulated.
Response: We have reformulated the part to “excessive SOD1 activity derived from the transgene”.
Line 314: It should read ‘Collectively, this evidence suggests that’.
Response: We have corrected it.
Line 315: It should read ‘cause for the emergence’.
Response: We have corrected it.
Line 328: It should read ‘antioxidative properties’.
Response: We have corrected it.
Line 331: It should read ‘caloric restriction’.
Response: We have corrected it.
Line 345: ‘toxins’ is unclear. Some detail must be provided as to what compounds are meant here.
Response: We have added “pertussis toxin”.
Line 353: It should read ‘vascular dysfunction compared to control mice [85].’.
Response: We have corrected it.
Line 368: ‘maintenance of lung homeostasis’ could be better formulated.
Response: We have reformulated it.
Line 383: It should read ‘In an initial study’.
Response: We have corrected it.
Line 386: ‘potential roles of the gene; should be reformulated.
Response: We have reformulated it.
Line 403: Instead of ‘to permissive levels’ could be replaced by ‘to manageable levels’.
Response: We have replaced it ‘to manageable levels’.
Line 406: It should read ‘in peroxisomes of’.
Response: We have corrected it.
Line 433: Given the important of Cys and GSH in the remainder of the manuscript it would be helpful to the reader if the range of free Cys and GSH in mammalian cells (e.g. hepatocytes) could be provided.
Response: Because amino acids and GSH concentrations vary depending on physiological state of cells, we provided approximate concentrations of them.
Line 437: It should read ‘, but does not cause a decrease in’.
Response: We have corrected it.
Line 438: ‘when used for a short time’ is unclear. Do the authors mean as a dietary supplement?
Response: We have rephrased the sentence to”during the short-term use of the inhibitor” to clarify the meaning.
Line 445: ‘other compounds’ is unclear and more detail about these must be provided.
Response: We have examples of ‘other compounds’.
Line 462: It would be helpful to the reader if the conc. range of Glu in the cell cytosol could be provided.
Response: Because amino acids concentrations vary depending on physiological state of cells, we provided approximate concentration.
Line 476: Remove the comma after GPX8.
Response: We have corrected it.
Line 482: ‘in such deaths’ is not a good formulation. “in this instance’ may be a better formulation.
Response: We have corrected it.
Line 490: It should read ‘GPX family is provided in a recent review article [130]’.
Response: We have corrected it.
Line 495: the abbreviation ‘PRDX’ must be explained.
Response: Full name with brief explanation of PRDX has been described in lines 170-171.
Line 500: ‘including chaperone activity’ please explain for which metal in particular?
Response: There is no relation with metal. We have added few words to explain chaperone activity of PRDX in protein folding.
Line 503: ‘in another partner’ is unclear and must be reformulated.
Response: We have clarified it.
Line 532: It should read ‘Other mechanisms of non-apoptotic cell death are classified’.
Response: We have corrected it.
Line 546: It should read ‘in the literature’.
Response: We have corrected it.
Line 552: ‘are self-divided’ is not a good formulation. Could it read ‘cells disintegrate into small’.
Response: We have corrected it.
Line 559: Remove ‘which’ and it should read ‘also appear to largely’.
Response: We have corrected it.
Line 564+5: It should read ‘examined in great detail.’ And ‘phospholipid that interacts’.
Response: We have corrected it.
Line 574: ‘performs catalysis’ please add of what?
Response: We have changed it to “perform hydrolysis of target proteins”.
Line 578: ‘L/S’ is unclear and needs to be explained. The sentence from his line to line 580 is unclear and needs to be reformulated.
Response: Cathepsin L/S, like cathepsin B, is the name of the enzyme, so no further explanation has been added.
Line 619: It should read ‘under Cys deficiency [156].’. To improve clarity it would be great if the cytosolic conc. of free Cys could be provided here.
Response: We have corrected it. There is no data on free Cys in the paper, we could not provide it here.
Line 621: It should read ‘deficiency’. Can details be provided with regard to what constitutes Cys deficiency (cytosolic Cys conc. < 0.05 mM)?
Response: Because each cell shows different sensitivity to Cys concentrations, it cannot be stated unconditionally.
Line 623: It should read ‘represented’.
Response: We have corrected it.
Line 629: It should read ‘of Cys deficiency’.
Response: We have corrected it.
Line 632: It should read ‘lysosome is the organelle’.
Response: We have added “the”.
Line 633: Should it read ‘than other subcellular compartments’ instead?
Response: We have corrected it.
Line 636: It should read ‘ferrous’.
Response: We have corrected it.
Line 637: It should read ‘out of the autolysosome’.
Response: We have corrected it.
Line 638: It should read ‘fragile which may potentially result in its rupture’.
Response: We have corrected it.
Lien 640: It should read ‘the integrity of the cell membrane’.
Response: We have corrected it.
Line 656: It should read “Proposed role of’.
Response: We have corrected it.
Line 664: It should read ‘have been recently reported’.
Response: We have corrected it.
Line 668: It should read ‘which initiate ferroptosis’.
Response: We have corrected it.
Line 671: It should read ‘Cys deficiency’.
Response: We have corrected it.
Line 675: It should read ‘studies reportedly mitigated ferroptosis’.
Response: We have corrected it.
Line 702: It should read ‘play a role in the’.
Response: We have corrected it.
Line 715: After ‘clusters’ the authors need to clarify from where to where these cluster proteins are transported.
Response: The issue is explained in the next sentence.
Line 719: Please provide details (structure) of sulfasalazine for context.
Response: We have added “, an inhibitor for xCT, ” to explain function of sulfasalazine.
Line 730: It should read ‘Figure 3 depicts a role of lipid’ and the abbreviation ‘ALOX’ must be explained.
Response: We have corrected the first issue and added explanation of ‘ALOX’ and moved the sentence to top of the next section.
Line 748: Detail of ferrostatin1 need to be provided for context.
Response: This sentence actually explains how vitamin E, ferrostatin-1, or enzymes, such as GPX4, inhibit lipid peroxidation and rescue cells from ferroptosis. So further explanation on ferrostatin-1 alone is redundant.
Line 761: Please remove comma after (FSP1).
Response: We believe this comma is essential because “designated as the ferroptosis-suppressor protein 1 (FSP1)” details “A redox protein”. So we changed the description to avoid misreading.
Lien 764; It should read ‘peroxidize PUFA’.
Response: We have corrected it.
Line 766+7: It should read ‘An inactivation/inhibition of GPX4’.
Response: We have corrected it.
Line 777: It should read ‘induce membrane destruction’.
Response: We have corrected it.
Line 807: It should read ‘in the regulation of’.
Response: We have corrected it.
Lien 810: ‘barely share responsible genes’ should be reformulated.
Response: We have rephrased it.
Line 811: It should read ‘effectively induces’.
Response: We have corrected it.
Line 816: It should read ‘p53 resulting in the stimulation’.
Response: We have corrected it.
Lien 865: ‘that are naturally present in our bodies’ should be changed to denote that they have to be ingested (one of the tocopherols is vitamin E, which needs to be orally ingested).
Response: We have corrected it.
Line 873: ‘to overcome this issue’ it is unclear which issue the authors refer to. Please specify.
Response: We have specifically described “the issue” as “solubility”.
Line 883: It should read ‘of the body.’.
Response: We have corrected it.
Line 900: It should read ‘in double’.
Response: We have corrected it.
Line 901: Please specify if you mean SOD1 and SOD3 specifically.
Response: We have described Akrla and SOD1, which were mentioned above, but repeated.
Line 920: ‘from the standpoint of anti-oxidation’ is not a good formulation. Please reformulate.
Response: We have rephrased it.
Line 950: ‘ONOO- is toxic to cells’ is not a good formulation as the dose makes the poison. Please provide context (i.e. is toxic at rather low doses of xyz).
Response: Because toxic doses vary depending on conditions, it is not possible to specify concentrations. No paper could determine toxic concentrations so far. So we have added description “when produced abundantly beyond antioxidative capacity”.
Line 961: Please remove the underlining of LDL.
Response: We have corrected it.
Line 1005: ‘with active metabolism’ is not a good formulation.
Response: We have reformulated it.
Line 1010+11: Please remove.
Response: We have removed two lines of description.
Reviewer 3 Report
Fujii and colleagues have collected the contribute of a large number of authors for what concern the involvement of superoxide radicals to the different cell death processes, particularly focusing on ferroptosis. In this review the state of the art of different cellular mechanisms involving the formation of radicals and their implication on cellular destiny is well explained and clearly organized. It is also appreciable how they emphasize which parts are still unknown.
The text reading is fluid and only some misprint have been detected. Some suggestion about the main errors are reported below:
page 1 line 14: has attracted
page 3 line 78 tricarboxylic acid
page 4 line 134: that DEphosphorylate
page 4 line 157: Cys-sulfornic acid
page 16 line 730: Figure "4"
page 17 line 792: convert "PFUA" in PUFA
page 18 line 818: ferrroptosis
page 22 line 1010-1011: to be removed
Author Response
We appreciate the reviewers' evaluation of the manuscript and their advice. We have amended it mostly according to your comments and advice.
page 1 line 14: has attracted
Response: We have corrected it.
page 3 line 78 tricarboxylic acid
Response: We have corrected it.
page 4 line 134: that Dephosphorylate
Response: We have corrected it.
page 4 line 157: Cys-sulfornic acid
Response: We have corrected it.
page 16 line 730: Figure "4"
Response: We have corrected it according to the comment and moved the sentence to top of the next section.
page 17 line 792: convert "PFUA" in PUFA
Response: We have corrected it.
page 18 line 818: ferrroptosis
Response: We have corrected it.
page 22 line 1010-1011: to be removed
Response: We have corrected it.